# Adapting Chat Language Models Using Only Target Unlabeled Language Data

**Atsuki Yamaguchi**                                             *ayamaguchi1@sheffield.ac.uk*
*University of Sheffield*

**Terufumi Morishita**
*Hitachi, Ltd.*

**Aline Villavicencio**                                          *a.villavicencio@sheffield.ac.uk*
*University of Exeter*
*University of Sheffield*

**Nikolaos Aletras**                                             *n.aletras@sheffield.ac.uk*
*University of Sheffield*

**Reviewed on OpenReview:** *https://openreview.net/forum?id=6IdoIKowfe*

## Abstract

Vocabulary expansion (VE) is the de-facto approach to language adaptation of large language models (LLMs) by adding new tokens and continuing pre-training on target data. While this is effective for base models trained on unlabeled data, it poses challenges for *chat* models trained to follow instructions through labeled conversation data. Directly adapting the latter with VE on target unlabeled data may result in forgetting chat abilities. While ideal, target chat data is often unavailable or costly to create for low-resource languages, and machine-translated alternatives are not always effective. To address this issue, previous work proposed using a base and chat model from the same family. This method first adapts the base LLM with VE on target unlabeled data and then converts it to a chat model by adding a chat vector (CV) derived from the weight difference between the source base and chat models. We propose ElChat, a new language adaptation method for chat LLMs that adapts a chat model directly on target unlabeled data, *without a base model*. It elicits chat abilities by injecting information from the source chat model. ElChat offers more robust and competitive target language and safety performance while achieving superior English, chat, and instruction-following abilities compared to CV.[1]

## 1 Introduction

Vocabulary expansion (VE) is the de-facto approach to adapting large language models (LLMs) in a target language (Cui et al., 2024; Fujii et al., 2024; Choi et al., 2024). It typically consists of two main steps: (i) new tokens are added to the model vocabulary by expanding the input embedding and output head matrices; and (ii) continual pre-training on target data to learn the input and output embeddings of the new tokens (Cui et al., 2024; Fujii et al., 2024; Choi et al., 2024; Tejaswi et al., 2024; Mundra et al., 2024, *inter alia*). VE is important because LLMs including chat models often perform poorly in languages underrepresented in the training data (Geng et al., 2025; Huang et al., 2024). Moreover, target language tokenization suffers from overfragmentation due to the heavy reliance on data and vocabulary from particular languages (e.g.,

---

[1] Our code is available on GitHub. The adapted models are available on Hugging Face Hub.

English), resulting to more inference steps especially in low-resource languages (Ahia et al., 2023; Petrov et al., 2023; Ali et al., 2024). Consequently, VE is a necessary step to mitigate this issue and achieve crucial inference speedups. While the vocabularies of frontier LLMs are often large, e.g., 152K for Qwen2.5 (Yang et al., 2024) and 128K for Llama 3.1 (Dubey et al., 2024), they still suffer from this overfragmentation in underrepresented languages. This means such languages often require substantially more inference steps than their high-resource counterparts. For example, processing a text sequence in Amharic requires 3.48x more inference steps using the default Qwen2.5 without VE.

While VE is effective for *base* models trained on unlabeled data, its application poses significant challenges when the LLM at hand is a *chat* model trained to follow instructions through labeled conversation data. Ideally, we need access to target chat data to effectively adapt chat models. However, this is often unavailable or costly to create for low-resource languages, including the acquisition of human feedback (Huang et al., 2024). Alternatively, machine-translated chat data are not consistently effective (Tao et al., 2024).

To address this issue, Huang et al. (2024) proposed chat vector (CV), a method to obtain a chat model in the target language with access to target unlabeled data only. CV first adapts the base LLM with VE on target unlabeled data and then converts it to a chat model by adding a chat vector derived from the weight difference between the source base and chat models. However, this requires access to base and chat models from the same family that might not always be available, hindering its applicability. For example, the Phi-3 (Abdin et al., 2024a) and Phi-4 (Abdin et al., 2024b) do not provide base models due to safety reasons.[2] Similarly, Velvet[3], EXAONE-3.5[4], and Trillion (Han et al., 2025b) models are available only as a chat model. Crucially, it is completely to the discretion of developers to decide whether they publish both base and chat variants.

In this paper, we propose ElChat, a new language adaptation method for chat LLMs that adapts a chat model directly on target unlabeled data, eliminating the need for a base model (Figure 1). We hypothesize that direct adaptation of a source chat model with VE on target unlabeled data negatively impacts its chat and instruction-following abilities by altering its parametric knowledge. However, we posit that these can still be recovered. For this purpose, ElChat leverages information from the source chat model to elicit chat abilities through two key mechanisms. First, we employ model merging to integrate distinct parametric knowledge from the source and target models (Wortsman et al., 2022; Yadav et al., 2023; Yu et al., 2024; Goddard et al., 2024). We hypothesize that model merging helps restore the chat and instruction-following abilities of the source model while preserving the target language performance achieved by the target model. Second, we reuse the weights of special tokens from the source model. For example, tokens that mark the start of a conversation turn should be crucial for activating the instruction-following ability as they are used to structure raw input into chat format. However, direct adaptation on target unlabeled data may degrade their functionality as they are modified during VE. To mitigate this, we copy these token weights directly from the source model to the target model.

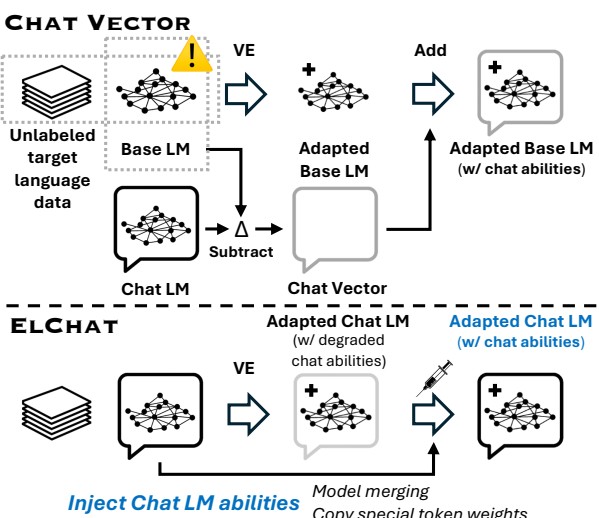

Figure 1: Chat LLM language adaptation with Chat Vector (Huang et al., 2024) and ElChat (ours). Note that a *base* LM in this paper refers to an LM pretrained on unlabeled data without any further post-training. A *chat* LM, on the other hand, is a base model that has been further supervised fine-tuned on labeled conversational data, enabling it to follow instructions.

---

[2] https://huggingface.co/microsoft/phi-4/discussions/4
[3] https://huggingface.co/Almawave/Velvet-14B
[4] https://huggingface.co/collections/LGAI-EXAONE/exaone-35-674d0e1bb3dcd2ab6f39dbb4

We investigate the efficacy of ElChat by experimenting with two popular chat models across seven typologically diverse languages. Our evaluation includes safety, chat, and instruction-following performance. Additionally, we also assess target and source language task performance and target language inference speed. Our key contributions are as follows:

- We propose ElChat that adapts a chat model directly on target unlabeled data, eliminating the need for (i) a base model and (ii) target chat data.

- ElChat achieves better chat and instruction-following abilities and source language performance than CV. It is also competitive and more robust (i.e., consistently outperforming the source chat model) in the target language and safety tasks compared to CV (§5.2, §5.3, §5.1).

- Despite model modifications, ElChat achieves comparable target inference speedups across models and tasks, matching the performance of the adapted VE and CV models (§6).

## 2 Related Work

### 2.1 Cross-lingual Vocabulary Adaptation

The most popular approach to adapting LLMs to a target language is by expanding their vocabulary (VE) with tokens from the target language (Balachandran, 2023; Larcher et al., 2023; Pipatanakul et al., 2023; Lin et al., 2024; Cui et al., 2024; Kim et al., 2024; Fujii et al., 2024; Choi et al., 2024; Nguyen et al., 2024; Tejaswi et al., 2024; Mundra et al., 2024).

Other methods to language adaptation include full or partial vocabulary replacement with a new target vocabulary (Ostendorff & Rehm, 2023; Csaki et al., 2023; Da Dalt et al., 2024; Remy et al., 2024; Yamaguchi et al., 2024a; Dobler & de Melo, 2024; Cahyawijaya et al., 2024), hypernetwork for tokenizer transfer (Minixhofer et al., 2024), and adapters for vocabulary alignment (Han et al., 2025a). Our work focuses on VE as it has been widely used recently for mostly base LLM adaptation in languages such as Chinese, Japanese, Korean, and Persian (Cui et al., 2024; Fujii et al., 2024; Choi et al., 2024; Mahdizadeh Sani et al., 2025, *inter alia.*).

### 2.2 Language Adaptation of Chat Models

Recent work has proposed developing chat models in a target language from source base models. For example, Toraman (2024) and Zhao et al. (2024a) apply VE to *base* models using target language chat data, consisting of 52k samples and 500M tokens, respectively. Bandarkar et al. (2025) also adapt base models using 30-40k target language chat data. Their approach also adapts a task-specific (i.e., math) model on 200k English math samples, followed by merging the two models to enhance math performance in the target language. Alexandrov et al. (2024) iteratively merge models trained on subsets of available target language data to effectively mitigate catastrophic forgetting. Their method first adapts base models with continual pre-training (CPT) on target unlabeled language data, followed by instruction tuning on target language chat data samples. However, this approach requires substantial target language data (at least 50B tokens of unlabeled data for CPT, and 78K samples of target language chat data) to ensure that each subset contains sufficient information for effective adaptation. A different approach, proposed by Tao et al. (2024), involves merging two base models: (1) one supervised fine-tuned on 162k English data samples, and (2) another trained on at least eight billion tokens of target unlabeled language data. However, it still relies on the availability of a base model. Geng et al. (2025) propose adapting source chat models directly through a multi-stage training approach. This method involves target unlabeled data and transfer fine-tuning (i.e., supervised fine-tuning tasks using translated target chat data.)

The main limitation with this line of work is that it requires access to target chat data (real or translated), typically in large volumes. Chat data is often unavailable or costly to produce for low-resource languages, while machine-translated chat data is not always effective for adaptation (Tao et al., 2024). For example, Burmese, one of our experimental languages, consists only of 472 manually annotated instruction samples in the Aya Dataset (Singh et al., 2024). This is insufficient for direct application of VE, as its typical data

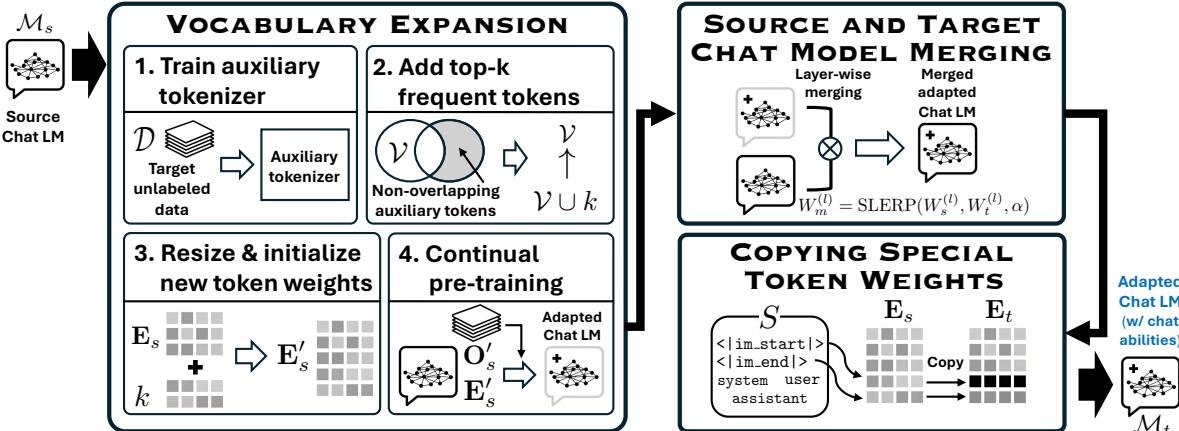

Figure 2: ElChat: A three-step adaptation process overview.

requirements are in the order of millions of tokens (Tejaswi et al., 2024), making these methods not applicable in such settings.

Huang et al. (2024) assumes a language adaptation setting where there is no access to target chat data (real or translated). This is a more realistic scenario for low-resource languages. For example, Burmese has only 172k target unlabeled language data in MADLAD-400 (Kudugunta et al., 2023) available for adaptation. Their proposed CV method obtains a target chat model using a source base and a source chat model. This approach to chat LLM adaptation is the closest to our work.

## 3 ElChat: Eliciting Chat and Instruction-following Abilities

Similar to Huang et al. (2024), we aim to adapt a chat LLM to a target language, assuming that we only have access to target unlabeled data and no target chat data (real or translated) at all. Unlike Huang et al. (2024), our goal is to remove the dependence on a base model that might not always be available and adapt a chat model directly on target unlabeled data with VE.

To achieve this, we introduce ElChat that consists of three steps: (i) VE on the source chat model using target unlabeled data to obtain an initial adapted target chat model; (ii) merging the source and target models; (iii) injecting information from the source to the target by copying tokens that are specific to chat and instruction-following capabilities. Figure 2 visualizes these steps. While the individual components of these are established techniques, the key novelty of ElChat resides in its strategic combination to overcome the dependence on the base model.

We expect that modifying the parametric knowledge of the source model by training on target unlabeled data in step (i) will improve its target language skills by updating specific areas of the network but negatively impact its chat and instruction-following abilities due to catastrophic forgetting. To remedy this, we hypothesize that we can elicit the latter through (ii) model merging (Wortsman et al., 2022; Yadav et al., 2023; Yu et al., 2024; Goddard et al., 2024). It allows the integration of distinct parametric knowledge from the source and target models without incurring any additional training costs, i.e., chat abilities from the source and target language knowledge from the target. Finally, in step (3), we restore the representation of special tokens used in the chat template. We assume this will guide the target model in effectively responding to user instructions by transferring this information from the source.

**Vocabulary Expansion (VE).**  Given the sole availability of target unlabeled data, we apply VE on the source chat LLM by expanding its input and output head matrices with new tokens, followed by CPT on the target data. This process follows the standard protocol adopted in Tejaswi et al. (2024) and Mundra et al. (2024) for resizing and initializing these matrices.

More specifically, given target unlabeled data $\mathcal{D}$ and a source chat model $\mathcal{M}_s$ with initial vocabulary $\mathcal{V}$:

1. An auxiliary target language tokenizer is first trained on $\mathcal{D}$. This tokenizer utilizes the same underlying tokenization function (e.g., byte-level BPE in this paper) as $\mathcal{M}_s$.

2. From this auxiliary tokenizer, we then select the top-$k$ most frequent tokens (e.g., $k = 10K$ by default, as in Tejaswi et al. (2024)) that do not overlap with $\mathcal{V}$ but are present in the auxiliary tokenizer's vocabulary. These selected $k$ tokens are then added to $\mathcal{V}$.

3. To accommodate these $k$ new tokens in $\mathcal{M}_s$, its input $\mathbf{E}_s \in \mathbb{R}^{|\mathcal{V}| \times H}$ and output head $\mathbf{O}_s \in \mathbb{R}^{H \times |\mathcal{V}|}$ matrices are resized. They become $\mathbf{E}'_s \in \mathbb{R}^{(|\mathcal{V}|+k) \times H}$ and $\mathbf{O}'_s \in \mathbb{R}^{H \times (|\mathcal{V}|+k)}$ respectively, where $H$ denotes the hidden dimensionality of $\mathcal{M}_s$. The weights corresponding to the newly added tokens are then initialized using mean initialization (Yao et al., 2021), a popular and simple, yet effective method in VE (Fujii et al., 2024; Tejaswi et al., 2024; Mundra et al., 2024). The weight of each newly added token is initialized as the average embedding or language modeling head weight of their corresponding source tokens, obtained using the corresponding source tokenizer.

After the above procedure, $\mathcal{M}_s$ undergoes continual pre-training on $\mathcal{D}$ using a causal language modeling objective. A key difference between chat and base models is that the former uses a chat template. This includes specific roles (e.g., user, system, or assistant) and has a placeholder for message text in the prompt (see Appendix A for details). For example, a chat template might structure input as follows:

```
<|im_start|>system
You are a helpful assistant.<|im_end|>
<|im_start|>user
What is the capital of the US?<|im_end|>
<|im_start|>assistant
```

During CPT, we remove the default chat template of the model to support unlabeled data because the unlabeled data typically lacks these explicit role annotations. During inference, we append it to task-specific prompt templates (see Table 3 for task-specific prompts).

**Source and Target Chat Model Merging (Merge).** After VE, we merge the source and target chat models. We employ a popular merging method: spherical linear interpolation (SLERP) (Goddard et al., 2024) to merge each layer of the source and adapted models.[5] This process excludes the embedding and language modeling head from merging because the source and target models use different vocabularies.

Given the source chat model $\mathcal{M}_s$ and the target-adapted chat model $\mathcal{M}_t$ (which has undergone VE), the weights of each corresponding layer $l$ (e.g., self-attention and feed-forward layers) are merged. For each such layer $l$, let $\mathbf{W}_s^{(l)}$ denote the weight matrix from $\mathcal{M}_s$, and $\mathbf{W}_t^{(l)}$ denote the corresponding weight matrix from $\mathcal{M}_t$. The merged weight matrix, denoted $\mathbf{W}_m^{(l)}$, is then computed using SLERP:

$$\mathbf{W}_m^{(l)} = \mathrm{SLERP}(\mathbf{W}_s^{(l)}, \mathbf{W}_t^{(l)}, \alpha)$$

where $\alpha \in [0, 1]$ is an interpolation coefficient. This $\alpha$ serves as a hyperparameter controlling the balance between $\mathcal{M}_s$ and $\mathcal{M}_t$ learned representations, where $\alpha = 0$ corresponds to using only the weights of $\mathcal{M}_s$ and $\alpha = 1$ to using only those of $\mathcal{M}_t$.

**Copying Special Token and Language Modeling Head Weights (Copy).** Special tokens used in a chat template (e.g., `<im_start>` in Qwen2.5 to represent the start of a turn) should be critical in supporting chat and instruction-following abilities of a model (see Appendix A for a full list of special tokens). Although the embedding and language modeling heads are excluded from merging due to vocabulary differences, leaving

---

[5]We also test linear merging (Wortsman et al., 2022) yielding similar results (see Appendix B). This suggests the robustness of our merging approach regardless of the specific interpolation method. Nonetheless, we primarily utilize SLERP to streamline our experiments due to its superior fine-grained control for merging, even though it may not be empirically superior across all tasks (Goddard et al., 2024).

them unchanged may not be optimal for eliciting the chat and instruction-following abilities of the adapted chat model. Hence, we copy all the special token weights from the source model to the adapted model.

Specifically, let $S$ be the set of token IDs corresponding to the special tokens defined by the chat template of $\mathcal{M}_s$. The input embedding matrix and the language modeling head of $\mathcal{M}_t$ are updated as follows:

For each special token ID $x \in S$, the embedding vector for token $x$ in $\mathcal{M}_t$, $\mathbf{E}_t \in \mathbb{R}^{(|\mathcal{V}|+k) \times H}$, is updated by copying the corresponding vector from $\mathcal{M}_s$'s input embedding matrix ($\mathbf{E}_s$): $\mathbf{E}_t[x,:] \leftarrow \mathbf{E}_s[x,:]$, where $\mathbf{E}[i,:]$ denotes the embedding vector for token ID $i$.

Similarly, the language modeling head weights corresponding to token $x$ in $\mathcal{M}_t$, $\mathbf{O}_t \in \mathbb{R}^{H \times (|\mathcal{V}|+k)}$, are updated by copying the corresponding vector from $\mathcal{M}_s$: $\mathbf{O}_t[:,x] \leftarrow \mathbf{O}_s[:,x]$, where $\mathbf{O}[:,j]$ denotes the column of weights for token ID $j$ in the language modeling head.

# 4 Experimental Setup

This section describes the experimental setup in this paper. More details are listed in Appendix A.

## 4.1 Source Models

We use two popular chat models as source: Qwen2.5 7B (Yang et al., 2024); and Llama 3.1 8B (Dubey et al., 2024), across experiments. Additionally, to ensure consistency in our analysis, we also incorporate the state-of-the-art chat model Qwen3 14B (Yang et al., 2025). The corresponding results and analysis for Qwen3 are presented in Appendix C.3.

## 4.2 Target Languages and Adaptation Data

We experiment with the following seven typologically diverse languages, assuming that they are likely to be underrepresented compared to English in the pre-training data of the source models, or entirely absent: Amharic (Afroasiatic), Bengali (Indo-European), Burmese (Sino-Tibetan), Gujarati (Indo-European), Sinhala (Indo-European), Tamil (Dravidian), and Telugu (Dravidian). The ratio of training data in each model for each source base and chat model has not been explicitly disclosed (Appendix C.2). Note that we do not consider Latin script target languages as they are less likely to suffer from overfragmentation and usually benefit less from VE (Yamaguchi et al., 2024b; Tejaswi et al., 2024). Due to computational constraints, we use Qwen3 for Amharic, Bengali, and Telugu only.[6]

For the CPT part of VE, we use MADLAD-400 (Kudugunta et al., 2023), which consists of highly-filtered document-level samples sourced from CommonCrawl, and randomly sample 250K language-specific documents for each language as the target unlabeled data.[7]

## 4.3 Continual Pre-training

Following Remy et al. (2024), we train the embedding, LM head, and the top and bottom two layers of a source model. This approach aims to calibrate only the parts closely related to the encoding and decoding of the target language (Wendler et al., 2024; Tang et al., 2024; Zhao et al., 2024b), minimizing changes to the source model while allowing cost-effective tuning.

## 4.4 Baselines

We compare ElChat against the following baselines:

- Off-the-shelf base (**Base**) and chat (**Chat**) models without target language adaptation.

---

[6] Amharic is selected because it shows the most significant speedup gains in target language generative tasks with Qwen2.5 and Llama 3.1 (Tables 17 and 18). Bengali and Telugu are chosen as they are the only languages covered by MGSM.

[7] For languages with less than 250K documents (i.e., Amharic and Burmese), we use the full articles.

- Base and Chat models adapted using standard VE, denoted by **Base+VE** and **Chat+VE** respectively. Note that the latter uses a chat template in inference (see §3).

- **CV** proposed by Huang et al. (2024), augmenting Base+VE with chat vector using Base and Chat.

For reference, we also experiment with adapting Chat and Base using only CPT on the same target language data without VE (i.e., no inference speedup in a target language). We provide the results and analysis of these CPT-only models in Appendix C.1.

### 4.5 Evaluation Tasks

We evaluate the efficacy of ElChat in safety, chat, and instruction-following performance, and target and source language performance.

**Safety, Chat, and Instruction-Following.**   Following Cahyawijaya et al. (2024), we conduct safety evaluation on target language translated data including TRUTHFULQA (Lin et al., 2022), TOXICGEN (Hartvigsen et al., 2022), and IMPLICITHATE (ElSherief et al., 2021). We also measure chat and instruction-following abilities in the source language (English) using IFEVAL (Zhou et al., 2023), GSM8K (Cobbe et al., 2021) as multi-turn few-shot, and MT-BENCH (Zheng et al., 2023). Furthermore, we measure the performance on English ALPACAEVAL v2.0 (Li et al., 2023; Dubois et al., 2024) for additional analysis.

Target language evaluation is challenging for instruction-following and chat tasks due to the limited data availability. LLM-as-a-Judge (Zheng et al., 2023) is also unstable according to Azime et al. (2024) in low-resource languages. Hence, we use multi-turn MGSM (Shi et al., 2023) for target language evaluation as it consists of manually translated high-quality data.

**Target Language.**   We use both generative and discriminative target language tasks. For generative tasks, we use summarization (SUM) using XL-SUM (Hasan et al., 2021) and English-to-target machine translation (MT) using FLORES-200 (NLLB Team et al., 2022). For a discriminative task, we employ multiple-choice text classification (MC) using Belebele (Bandarkar et al., 2024) and Global MMLU (GMMLU) (Singh et al., 2025) as general target language understanding benchmarks.

**Source Language (English).**   We assess the extent to which the adapted models retain their general task-solving abilities in English SUM, target-to-English MT, and English MC using the same datasets as those employed for target languages. We also use MMLU (Hendrycks et al., 2021) as an English language understanding benchmark and English BBH (Srivastava et al., 2023; Suzgun et al., 2023) as a stress-test benchmark.

Following Ahia et al. (2023), we use 500 random samples for generative tasks: SUM and MT. The rest use the full test sets for evaluation.

### 4.6 Evaluation Metrics

**Task Performance.**   We report the standard metrics for each task: accuracy for MC, GMMLU, MMLU, BBH, TRUTHFULQA, and IFEVAL (strict prompt), and exact match for GSM8K and MGSM. For MT-BENCH, we use the mean score over two turns across all questions. Adhering to the standard protocol in LightEval (Fourrier et al., 2023), each score is determined using `Flow-Judge-v0.1` and follows a Likert-5 scale. For ALPACAEVAL, we use a win-rate over GPT-4 (1106 Preview) measured by GPT-4.1 nano (2025-04-14).[8] For SUM and MT, we primarily use chrF (Popović, 2015).[9] For TOXICGEN and IMPLICITHATE, we use safety score, which is the percentage of likeliness of the model producing benign over harmful sentences, following Cahyawijaya et al. (2024).

---

[8] https://openai.com/index/gpt-4-1/
[9] Although chrF has been a widely used metric for SUM and MT (Ebrahimi et al., 2023; Remy et al., 2024, *inter alia*), we also show ROUGE-L (Lin, 2004) for SUM and chrF++ (Popović, 2017) for MT in Appendix B.

We report average zero- and three-shot performance across three different runs for SUM and MT, respectively. For the remaining tasks, we report single-run zero-shot performance for IFEVAL, MT-BENCH, TOXIC-GEN and IMPLICITHATE, three-shot performance for MC, TRUTHFULQA, five-shot performance for GMMLU, MMLU, GSM8K, and MGSM as these tasks are deterministically evaluated with temperature set to zero.

**Inference Efficiency.** VE offers inference speedups in a target language compared to source models (Tejaswi et al., 2024; Mundra et al., 2024; Yamaguchi et al., 2024b). To quantify this, we measure the number of tokens generated per second (tokens/s) (Hong et al., 2024).

## 5 Task Performance

### 5.1 Safety, Chat, and Instruction-following

**Safety.** Figure 3 (a) shows the aggregated mean performance in safety tasks across the seven target languages. We first observe that ElChat outperforms CV in TRUTHFULQA for both Qwen2.5 and Llama 3.1. In particular, it substantially helps Llama 3.1, achieving 13-point gains over CV on average. We speculate that CV may be less effective for classification tasks as reflected in its performance on other discriminative target language tasks (§5.2).

In contrast, CV often surpasses ElChat in TOXI-GEN and IMPLICITHATE for both models, with differences of up to 6 points, e.g., in TOXIGEN with Qwen2.5. This is primarily due to the use of Merge instead of Copy in ElChat (see Table 7). Specifically, ElChat without Copy follows similar trends to ElChat, while ElChat without Merge exhibits similar trends to Chat+VE and CV. This is intuitive, as merging a model with the lowest target safety performance (i.e., Chat) with Chat+VE can degrade safety performance.

We finally find that ElChat, CV, and Chat+VE outperform the Chat baseline across tasks and models,

(a) Target Language

(b) English (Source)

Figure 3: Aggregated mean performance (↑) across seven target languages for each model on safety, chat, and instruction-following tasks. Full results are in the Appendix (Tables 6 and 7).

with gains ranging from 1.4 (TRUTHFULQA with Llama 3.1 using ElChat) to 20 points (TRUTHFULQA with Qwen2.5 using ElChat). The only exceptions are CV and Chat+VE in TRUTHFULQA, where they underperform Chat by 11 and 14 points, respectively. This instability highlights the advantage of ElChat, as it consistently enhances safety performance over Chat across tasks.

**Chat and Instruction-following.** Figure 3 (b) shows the aggregated mean performance across chat and instruction-following tasks in English (source). We first analyze the extent to which ElChat impacts performance on chat and instruction-following tasks compared to the Chat baseline. Note that some performance degradation is inevitable, as adapting Chat to *target* unlabeled data inherently affects the *source* chat and instruction-following abilities (§3).

As anticipated, we find that ElChat exhibits performance degradation across tasks and models compared to the Chat baseline. However, the extent of this degradation varies depending on the model, task, and adaptation approach. For example, ElChat achieves comparable performance on GSM8K but experiences reductions of 16 and 0.57 points on IFEVAL and MT-BENCH, respectively. Despite these drops, ElChat successfully improves instruction-following performance compared to the respective adapted model, Chat+VE. It demonstrates improvements of 14 and 13 points over Chat+VE for Qwen2.5 and Llama 3.1, respectively. These results indicate that ElChat can inject instruction-following capabilities into the adapted model.

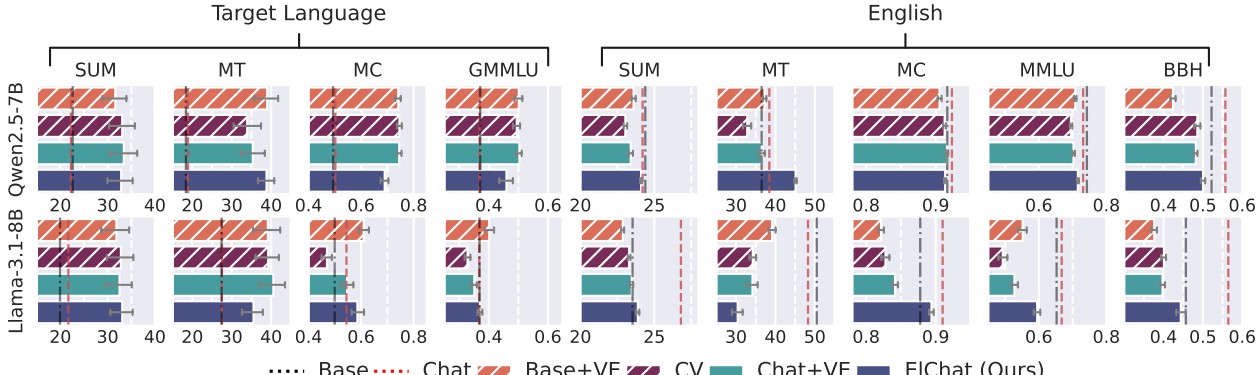

Figure 4: Aggregated mean performance across seven target languages for each model (error bars indicate 95% confidence interval). Full results are in Tables 11 and 12.

We next observe that ElChat generally outperforms CV across tasks and models in five out of six cases, with performance differences ranging from 9.5 points (IFEVAL with Llama 3.1) to 25 points (GSM8K with Llama 3.1). A similar trend is observed on ALPACAEVAL (Table 10 in the Appendix), where ElChat substantially outperforms CV with up to a 16-point gain in Qwen2.5, while still performing competitively with only a 0.95-point drop in Llama 3.1. These results suggest that ElChat is more effective than CV in enhancing both chat and instruction-following abilities.

Finally, ElChat's performance advantage extends to target language tasks. Table 1 shows model performance in MGSM (covering Bengali and Telugu). Overall, ElChat surpasses both CV and the original chat model, Chat, in three and four out of four cases, respectively. Notably, ElChat significantly improves Telugu performance by 29 points with both Qwen2.5 and Llama 3.1. Additionally, while CV enhances Bengali performance by just 1 point over Chat in Llama 3.1, ElChat achieves a substantial 21-point gain. These results further support the superiority of ElChat over CV.

Table 1: MGSM performance in Bengali (bn) and Telugu (te) by model.

(a) Qwen2.5 7B

| Model | EM | |
|---|---|---|
| | bn | te |
| Chat | .23 | .06 |
| CV | **.60** | .27 |
| Chat+VE | .39 | .27 |
| ElChat (Ours) | .46 | **.35** |

(b) Llama 3.1 8B

| Model | EM | |
|---|---|---|
| | bn | te |
| Chat | .30 | .12 |
| CV | .31 | .24 |
| Chat+VE | .26 | .28 |
| ElChat (Ours) | **.51** | **.41** |

## 5.2 Target Language

Figure 4 (left) shows the aggregated mean performance across seven languages for all source and adapted target models in target language.

We note that ElChat consistently outperforms its source chat model (Chat) across all models and tasks. This improvement is particularly notable in generative tasks (i.e., SUM and MT), with gains ranging from 8 points (Llama 3.1 on MT) to 20 points (Qwen2.5 on MT). While ElChat generally maintains competitive performance (within 3 points) compared to Chat+VE, it exhibits slightly reduced performance (up to 5.9 points with Qwen2.5 on MC) in discriminative tasks (i.e., MC and GMMLU) and MT with Llama 3.1. These results suggest that ElChat can overall preserve the target language performance, while the model modifications do not substantially degrade performance.

We further observe that ElChat demonstrates competitive performance with CV, with each method outperforming the other in half of the evaluated cases. Specifically, CV generally outperforms ElChat in Qwen2.5, except for the MT task, whereas ElChat typically achieves better performance than CV in Llama 3.1, excluding MT. However, CV notably underperforms both the source base and chat models in the two discriminative tasks (i.e., MC and GMMLU) with Llama 3.1. Thus, although ElChat and CV achieve similar overall performance, our method is more likely to yield improvements in target language tasks.

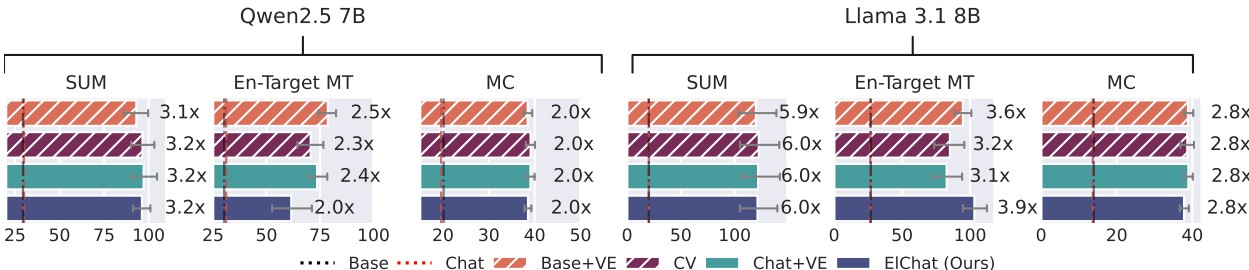

Figure 5: Aggregated mean inference speedup (tokens/s) across seven target languages. The value next to each bar represents the speedup ratio over Chat. Full results are in the Appendix (Tables 17 and 18).

### 5.3 Source Language (English)

Figure 4 (right) shows the aggregated mean performance across seven languages for all source and adapted target models in English tasks. ElChat generally underperforms the source Chat baseline. However, the degree of performance degradation varies considerably depending on the task and model similar to §5.1. For instance, with Llama 3.1, ElChat exhibits substantial performance drops of 7, 12, and 18 points on MMLU, BBH and MT, respectively. In contrast, the performance degradation observed with Qwen2.5 is less pronounced, with a maximum decrease of 5.7 points on BBH. Interestingly, ElChat even demonstrates a 6.7-point improvement on MT with Qwen2.5. This improvement likely stems from two key factors: (1) ElChat's effective utilization of source tokens (95%) during generation, compared to other approaches (i.e., Base+VE, CV, and Chat+VE) that achieve at most 71% source token utilization; and (2) its successful early stopping, generating a similar number of tokens (33) as Chat (see Figures 7 and 8 in the Appendix).

Comparing ElChat and Chat+VE, we find that ElChat generally yields better or comparable performance across models and tasks, with the exception of MT with Llama 3.1. This improvement suggests that ElChat can generally alleviate catastrophic forgetting not only in chat and instruction-following tasks (§5.1) but also in source language tasks by injecting the source chat information into Chat+VE.

Conversely, CV exhibits poor performance on English tasks, typically falling short of ElChat in eight out of ten cases. Moreover, the benefits of catastrophic forgetting mitigation are not consistently observed with CV, as it only outperforms its adapted base model, Base+VE, in five out of ten cases. This result somewhat contrasts with Huang et al. (2024), who highlight the benefits of using CV to mitigate catastrophic forgetting and improve knowledge retention and language ability. We speculate that modifying models through simple arithmetic operations, as in CV, may be less robust than our method. These results suggest that ElChat more effectively integrates target language abilities while mitigating performance degradation across chat, instruction-following, and source English tasks compared to CV.

## 6 Inference Efficiency

VE offers inference speedups on target language tasks by reducing text overfragmentation (§4.6). Since ElChat and CV involve modifications to model weights, they may increase the generation of source tokens, potentially slowing inference compared to Chat+VE and Base+VE. Furthermore, the inference efficiency of CV with VE has not been previously studied. Therefore, we investigate the inference efficiency of both methods. Figure 5 shows the aggregated mean inference speedups across seven languages and three target tasks.

Overall, both ElChat and CV achieve comparable speedups to their respective adapted models, Chat+VE and Base+VE, demonstrating 2.0x to 6.0x speedups over the source chat model, Chat. The largest drop is observed between ElChat and Chat+VE on the MT task with Qwen2.5 (0.4x). This likely happens because ElChat frequently terminates generation early (see Table 8), resulting in fewer generated tokens (and thus a smaller numerator in the tokens/second calculation). In summary, these results suggest that the model

[User] టోలా ఒక బేకరీకి వెళ్లి, వివిధ రకాలైన పేస్ట్రీలను కొనుగోలు చేసింది. ఆమె డజను $68 ఖరీదుతో 3 డజన్ల డోనట్లను, డజను $80 డాలర్లు ఖరీదుతో 2 డజన్ల మినీ కప్ కేకులను, డజను $55 ఖరీదుతో 6 డజన్ల మినీ చీజ్‌కేక్‌లను కొనుగోలు చేసింది. మొత్తం ఖర్చు ఎంత?

*Toula went to a bakery and bought a variety of pastries. She bought 3 dozen donuts for $68 a dozen, 2 dozen mini cupcakes for $80 a dozen, and 6 dozen mini cheesecakes for $55 a dozen. What was the total cost?*

→[ElChat] మినీ చీజ్‌కేక్‌లు 6*55=330 డాలర్లు ఖరీదుతో 6 డజన్ల కొనుగోలు చేయబడ్డాయి. మినీ కప్ కేకులు 2*80=160 డాలర్లు ఖరీదుతో 2 డజన్ల కొనుగోలు చేయబడ్డాయి. డోనట్లు 3*68=204 డాలర్లు ఖరీదుతో 3 డజన్ల కొనుగోలు చేయబడ్డాయి. మొత్తం ఖర్చు 330+160+204=694 డాలర్లు. సమాధానం 694.

*Mini cheesecakes were purchased at a cost of 6*55=330 dollars, 6 dozen. Mini cupcakes were purchased at a cost of 2*80=160 dollars, 2 dozen. Donuts were purchased at a cost of 3*68=204 dollars, 3 dozen. Total cost was 330+160+204=694 dollars. Answer is 694.*

(a) Correct answer - Correct reasoning

[User] রায়ান তাঁর বাগানে একদিনে 2টি ফুল লাগিয়েছেন। 15 দিন পরে, যদি 5টি না বেড়ে ওঠে তবে মোট কটি ফুল থাকবে?

*Ryan planted 2 flowers in his garden in one day. After 15 days, if 5 do not grow, how many flowers will be left in total?*

→[ElChat] রায়ান তাঁর বাগানে 1 দিনে 2টি ফুল লাগিয়েছেন। 15 দিন পরে তিনি 2 * 15 = 30টি ফুল লাগিয়েছেন। তিনি 5টি ফুল লাগিয়েছেন না। তাই তাঁর বাগানে মোট 30 - 5 = 25টি ফুল থাকবে। উত্তর হল 25।

*Ryan planted 2 flowers in his garden in 1 day. After 15 days he planted 2 * 15 = 30 flowers. He did not plant 5 flowers. So there will be a total of 30 - 5 = 25 flowers in his garden. The answer is 25.*

(b) Correct answer - Wrong reasoning

Figure 6: Output examples on MGSM. We use Google Translate for translation. Few-shot demonstrations (conversation history) are omitted due to space constraints.

modifications introduced by ElChat and CV do not negatively impact the inference speedups provided by VE.

## 7 Analysis

To better understand the behavior of ElChat, we perform an ablation in chat and instruction-following tasks using Llama 3.1. We also conduct a qualitative analysis using MGSM to gain further insights.

**Ablation.** Table 2 shows the importance of each ElChat component. Removing either of the two components (§3) significantly reduces performance across chat and instruction-following tasks, demonstrating their complementarity in eliciting ElChat's abilities. For instance, ablating the weight copying (Copy) component, leaving only VE and model merging, causes performance to drop by an average of 8 points on IFEval, 21 points on GSM8K, 14 points on MGSM, and 0.27 points on MT-Bench

Table 2: Mean performance across languages for chat and instruction-following tasks using Llama 3.1. Ablation results in other tasks are available in Appendix B.

| Model | IFEval | GSM8K | MGSM | MT-Bench |
|---|---|---|---|---|
| CV | $.38_{03}$ | $.33_{11}$ | $.27_{03}$ | $3.04_{20}$ |
| ElChat | $.47_{04}$ | $.58_{06}$ | $.46_{05}$ | $2.92_{27}$ |
| – Merge | $.34_{05}$ | $.40_{18}$ | $.37_{06}$ | $2.53_{22}$ |
| – Copy | $.39_{02}$ | $.37_{17}$ | $.32_{04}$ | $2.65_{23}$ |

relative to ElChat. This substantially undermines the advantage of ElChat over CV as it greatly lags behind CV in MT-Bench and shows marginally better performance than CV with only a few points (up to 4 points on GSM8K) differences in other tasks, unlike at least 9 points difference observed between ElChat and CV. These results clearly demonstrate that while the individual components are established techniques, their synergistic effect within a chat model context is non-trivial, and their combination is essential for ElChat's strong performance.

**Qualitative Analysis.** Figure 6 presents examples of ElChat's output on MGSM, highlighting both successes and challenges. Case (a) showcases correct reasoning and answer generation in Telugu. However, case (b) demonstrates that even when providing a correct answer in Bengali, ElChat can exhibit wrong reasoning. The misinterpretation of "five flowers do not grow" as "Ryan did not plant five flowers" suggests a potential limitation in understanding nuanced language. Notably, in the same case, it correctly reasons in English that "5 did not grow". Focusing on improving target language abilities further during VE while mitigating catastrophic forgetting of chat and instruction-following could address this issue. For instance,

making iterative model merging (Alexandrov et al., 2024) applicable to low-resource settings is a potential avenue for future investigation.

## 8 Conclusion

We introduced ElChat, a method for directly adapting *chat* models with VE using unlabeled data, eliminating the need for a *base* model and target chat data. To mitigate potential catastrophic forgetting in the adapted chat models after VE, ElChat elicits chat abilities by injecting information from the source chat model without requiring further training. Extensive experiments across safety, chat, and instruction-following, target language, and source language tasks demonstrated that ElChat outperforms the previous state-of-the-art CV approach in chat and instruction-following, and English tasks while being competitive and more robust in the target language and safety tasks. These results highlight ElChat's superior abilities.

## Limitations

**Continual Pre-training Methods.** This paper uses a continual pre-training method proposed by Remy et al. (2024), which tunes the top and bottom two layers of a model and its embedding and language modeling head, for efficient and effective target language adaptation. Nonetheless, other continual pre-training methods exist, including adapter-based training (e.g., LoRA (Hu et al., 2022)) and full fine-tuning. It would be interesting to extensively investigate the effect of different training methods for future work, but this is beyond the scope of this paper.

**Model Merging Methods.** We experiment with linear and SLERP merging as representative model merging methods for simplicity. More recent methods like TIES (Yadav et al., 2023) and DARE-TIES (Yu et al., 2024) might perform even better in ElChat. Given the resource constraints, we leave this investigation for future work.

**Languages.** This paper covers seven typologically diverse languages, following previous work on language adaptation that has also tested a similar number of languages. For instance, Minixhofer et al. (2022) tested eight languages. Note that Huang et al. (2024) used three languages (i.e., Chinese, Traditional Chinese, and Korean) to verify the effectiveness of CV. Experimenting with more languages is an interesting avenue for future work but is out of the scope of this paper, given our limited computing capacity.

**Chat and Instruction-following Evaluation.** Our chat and instruction-following evaluation is mainly on English data except for MGSM due to the limited availability of manually curated language-specific evaluation resources. Azime et al. (2024) has also noted the instability of using LLM-as-a-Judge in Amharic, which is also one of our experimental languages. It would be an interesting avenue to explore more chat and instruction-following evaluation in target languages for future work. We hope our work inspires the development of extensive evaluation benchmarks in low-resource languages.

## Ethical Considerations

Although we conducted extensive experiments across diverse public datasets to validate the effectiveness of ElChat, these datasets do not fully represent all real-world scenarios. Therefore, any model derived from or based on this work should be used with caution.

While this work does not appear to raise immediate concerns, the deployment of adapted chat LMs, especially in under-monitored, low-resource language regions, warrants further analysis. These models could inadvertently perpetuate harmful biases, compromise safety in ways not captured by current benchmarks, or be exploited for misinformation and other malicious purposes. Further research and responsible deployment strategies are crucial to address these potential risks.

# Acknowledgments

We would like to thank the Action Editor, Ruoyu Sun, the TMLR reviewers, Xi Wang, Sam Lewis-Lim, Maggie Mi, Huiyin Xue, and Xingwei Tan for their valuable feedback. We acknowledge (1) IT Services at the University of Sheffield for the provision of services for high-performance computing; (2) the use of the University of Oxford Advanced Research Computing (ARC) facility; (3) EuroHPC Joint Undertaking for awarding us access to MeluXina at LuxProvide, Luxembourg; and (4) the use of resources provided by the Isambard-AI National AI Research Resource (AIRR). Isambard-AI is operated by the University of Bristol and is funded by the UK Government's Department for Science, Innovation and Technology (DSIT) via UK Research and Innovation; and the Science and Technology Facilities Council [ST/AIRR/I-A-I/1023]. AY is supported by the Engineering and Physical Sciences Research Council (EPSRC) [grant number EP/W524360/1] and the Japan Student Services Organization (JASSO) Student Exchange Support Program (Graduate Scholarship for Degree Seeking Students).

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

# A    Experimental Setup

## A.1    Chat Template and Special Tokens

Model-specific chat templates and special tokens are accessible via the following links:

- **Qwen2.5**: `https://huggingface.co/Qwen/Qwen2.5-7B-Instruct/blob/main/tokenizer_config.json`

- **Llama 3.1**: `https://huggingface.co/meta-llama/Llama-3.1-8B-Instruct/blob/main/tokenizer_config.json`

- **Qwen3**: `https://huggingface.co/Qwen/Qwen3-14B/blob/main/tokenizer_config.json`

Below are excerpts from the chat templates of each model with placeholders for a prompt and output:

**Qwen2.5.**

```
<|im_start|>system
You are Qwen, created by Alibaba Cloud.
You are a helpful assistant.
<|im_end|>
<|im_start|>user
{prompt}
<|im_end|>
<|im_start|>assistant
{output}
```

**Llama 3.1.**

```
<|begin_of_text|>
<|start_header_id|>system
<|end_header_id|>
Cutting Knowledge Date: December 2023
Today Date: 26 Jul 2024
<|eot_id|>
<|start_header_id|>user
<|end_header_id|>
{prompt}
<|eot_id|>
<|start_header_id|>assistant
<|end_header_id|>
{output}
```

**Qwen3.**

```
<|im_start|>user
{prompt}
<|im_end|>
<|im_start|>assistant
<think>
</think>
{output}
```

## A.2   Prompt Template

We translate the English prompt templates provided by Ahia et al. (2023) for SUM with a machine translation API, following Yong et al. (2023). For MT and MC, we formulate a task-specific English prompt, followed by machine translation for each language. For the remaining tasks, except for MT-BENCH, we use the default templates provided in `lm-evaluation-harness` (Gao et al., 2023). For MT-BENCH, we use the default template provided in LightEval (Fourrier et al., 2023). Table 3 shows the prompt templates used in our evaluation. Note that we do not make any changes to the task-specific prompt to allow for a fair comparison between models with and without a chat template (i.e., Base and Chat, respectively).

Table 4: Hyperparameters for continual pre-training.

| Hyperparameters | Values |
|---|---|
| Batch size | 32 |
| Number of training steps | 30,517 |
| Adam $\epsilon$ | 1e-8 |
| Adam $\beta_1$ | 0.9 |
| Adam $\beta_2$ | 0.999 |
| Sequence length | 512 |
| Learning rate | 5e-5 |
| Learning rate scheduler | cosine |
| Warmup steps | First 5% of steps |
| Weight decay | 0.01 |
| Attention dropout | 0.0 |
| Training precision | BF16 |

Table 5: Parameters for non-greedy generative tasks: MT and SUM.

| Parameters | Values |
|---|---|
| Temperature | 0.8 |
| Repetition penalty | 1.1 |
| Top $k$ | 40 |
| Top $p$ | 0.9 |
| Beam width | 5 |
| Sampling | True |
| Early stopping | True |
| Maximum number of generated tokens | 128 |

### A.3 Implementation Details

Our experimental design is based on the findings from Tejaswi et al. (2024). They report that (1) *there are no significant gains when adding more than 10K tokens to the source vocabulary*, and (2) *additional CPT in the order of millions of tokens is sufficient for model adaptation*. Given this, we set the vocabulary size of the auxiliary target language tokenizer $|\mathcal{V}_{\mathrm{aux}}|$ to 50K across languages and the number of new target tokens $k$ to 10K. We train each model for 500M tokens with a batch size of 32, a maximum learning rate of 5e-5, and a sequence length of 512.

**Tokenizer Training.**  We train tokenizers using Hugging Face Tokenizers.

**Preprocessing.**  We preprocess datasets with Hugging Face Datasets (Lhoest et al., 2021).

**Continual Pre-training.**  We implement our models using PyTorch (Ansel et al., 2024) and Hugging Face Transformers (Wolf et al., 2020). Table 4 lists the hyperparameters in CPT.

**Model Merging.**  To ensure a smooth transition between layers, we use a $0.3 : 0.7 =$ Chat : Chat+VE mixing ratio for the top and bottom layers of all merged models, favoring Chat+VE as these layers are adjacent to the embeddings and language modeling head of Chat+VE. For the second top and bottom layers, we use $0.5 : 0.5 =$ Chat : Chat+VE, balancing the contributions of Chat and Chat+VE. For Qwen3, we use a $0.1 : 0.9 =$ Chat : Chat+VE mixing ratio for all layers.

**Evaluation.** We use Hugging Face LightEval[10] for evaluation on all tasks except for ALPACAEVAL, IFE-VAL, GSM8K and MGSM. For ALPACAEVAL, we use the official implementation available on GitHub[11] (v0.6.6). For IFEVAL, GSM8K, and MGSM, we use `lm-evaluation-harness` (Gao et al., 2023). To compute ROUGE-L, we split sentences with an mT5 (Xue et al., 2021) tokenizer as preprocessing following Maynez et al. (2023) and subsequently call `rouge_scorer`[12] to compute the metric. To compute chrF and chrF++, we use SacreBLEU (Post, 2018). For safety evaluation, we follow Cahyawijaya et al. (2024) and use their implementation available on GitHub: https://github.com/IndoNLP/cendol.

Table 5 lists the parameters used during evaluation for generative tasks: MT and SUM. To make a fair comparison, we do not conduct any generation parameter tuning and use the same ones across all approaches.

**Hardware.** We use either a single NVIDIA A100 (80GB), NVIDIA H100 (80GB), or NVIDIA GH200 (96GB) for CPT. For CPT with Qwen3 14B, we use a single AMD MI300X GPU. For evaluation, we use a single NVIDIA A100 (80GB) for all Llama 3.1 models, a single NVIDIA H100 (80GB) for all Qwen2.5 models, and a single AMD MI300X GPU for all Qwen3 models to ensure accurate measurement of inference efficiency.

**Model Links.** We list all the source model URLs in the following:

- **Qwen2.5** (Chat): https://huggingface.co/Qwen/Qwen2.5-7B-Instruct

- **Qwen2.5** (Base): https://huggingface.co/Qwen/Qwen2.5-7B

- **Llama 3.1** (Chat): https://huggingface.co/meta-llama/Llama-3.1-8B-Instruct

- **Llama 3.1** (Base): https://huggingface.co/meta-llama/Llama-3.1-8B

- **Qwen3** (Base): https://huggingface.co/Qwen/Qwen3-14B-Base

- **Qwen3** (Chat): https://huggingface.co/Qwen/Qwen3-14B

## B  Results

### B.1  Task Performance

- **Safety, Chat, and Instruction-following**: Tables 6, 7, and 8 provide a detailed breakdown of the task performance results for Qwen2.5, Llama 3.1, and Qwen3 across safety, chat, and instruction-following tasks. Table 9 shows the results of Chat+CPT and ElChat-related ablation models in MGSM. Table 10 shows the results of Chat-related models on ALPACAEVAL.

- **Target Language and Source (English)**: Tables 11, 12, and 13 provide a detailed breakdown of the task performance results for Qwen2.5, Llama 3.1, and Qwen3 across target language and source (English) language tasks.

- **sum and mt Results with Auxiliary Metrics**: Tables 14, 15, and 16 provide a detailed breakdown of SUM and MT performance results, measured by ROUGE-L for SUM and chrF++ for MT, for Qwen2.5, Llama 3.1, and Qwen3, respectively.

### B.2  Inference Efficiency

Tables 17, 18, and 19 provide a detailed breakdown of the inference efficiency results for Qwen2.5, Llama 3.1, and Qwen3, respectively.

---

[10]https://github.com/huggingface/lighteval
[11]https://github.com/tatsu-lab/alpaca_eval
[12]https://github.com/google-research/google-research/tree/master/rouge

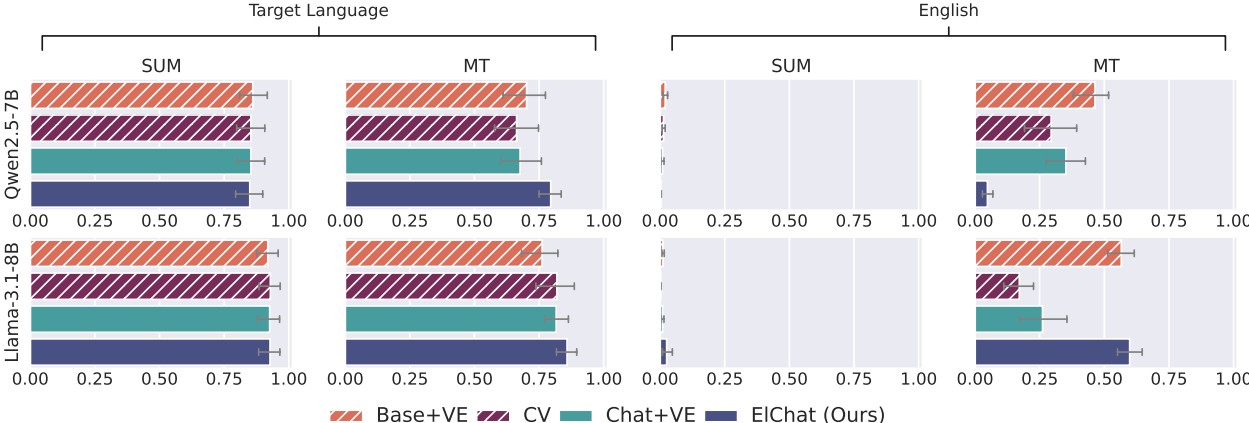

Figure 7: Aggregated average ratio of target tokens in output per sample across seven target languages for each model (error bars indicate 95% confidence interval).

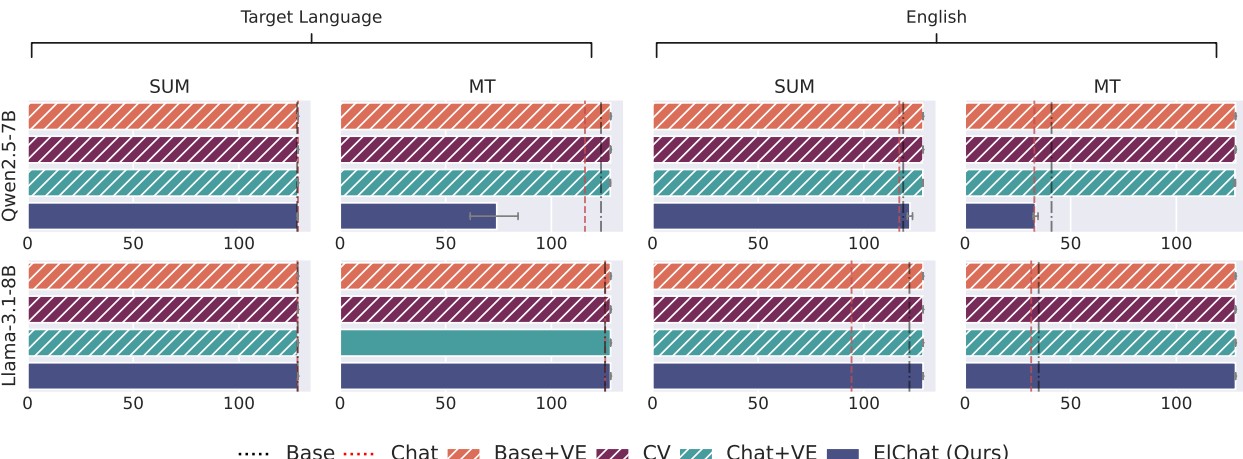

Figure 8: Aggregated average number of generated tokens per sample across seven target languages for each model (error bars indicate 95% confidence interval).

### B.3 Ratio of Target Language Tokens

Figure 7 shows the aggregated mean ratio of target new tokens in output per sample across seven target languages for each model.

### B.4 Number of Generated Tokens

Figure 8 shows the aggregated average number of generated tokens per sample across seven target languages for each model.

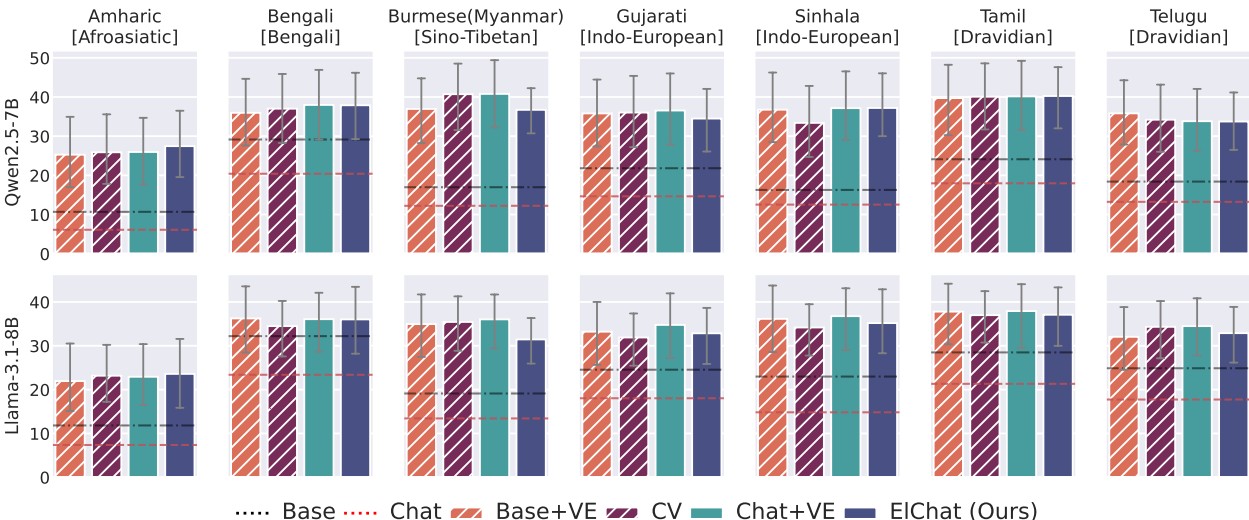

Figure 9: Aggregated mean performance across target language tasks for each model by language (error bars indicate 95% confidence interval).

# C   Analysis and Discussion

## C.1   CPT-only vs. VE

Comparing the adapted chat models (Chat+VE and ElChat) with the CPT-only model (Chat+CPT) in Tables 11 and 12, we observe three key trends: (1) Chat+VE generally performs better than Chat+CPT on target language tasks across both models; (2) Chat+CPT often outperforms Chat+VE on MT tasks; and (3) ElChat either matches or surpasses Chat+CPT on nearly all tasks and models, except for Target MC and GMMLU with Qwen2.5 and target-to-English MT with Llama 3.1. This performance advantage of ElChat is also confirmed for chat and instruction-following tasks (Tables 6, 7, and 9), where ElChat substantially outperforms Chat+CPT in almost all cases across languages, tasks, and models.

We also observe a similar trend between Base+VE and the CPT-only adapted base model (Base+CPT) in Tables 11 and 12: Base+VE often outperforms Base+CPT in most of the tasks, while Base+CPT excels in target-to-English MT across models.

These results somewhat contradict the recent observations (Downey et al., 2023; Yamaguchi et al., 2024a;b) that CPT-only models often perform better than vocabulary adapted models, possibly due to the robustly aligned original embeddings. However, Fujii et al. (2024) reported that "*the overall impact of vocabulary expansion on performance is minor.*" Further, Dobler & de Melo (2024) also claimed that "*we do not see a clear trend of better performance with or without tokenizer swapping*" for their vocabulary adaptation experiments. We hypothesize that the superiority of CPT is greatly affected by the amount of CPT data, and it can be more apparent in low-resource settings as in Yamaguchi et al. (2024b), where new embeddings are likely to be not well aligned.

It is important to note that the CPT-only models (i.e., Base+CPT and Chat+CPT) have no speedups at all (Tables 17 and 18) as they use the same vocabulary as the source models (i.e., Base and Chat).

## C.2   Additional Analysis by Language

We conduct additional analysis of the target language task performance of models by language. Figure 9 presents the aggregated mean performance across three target language tasks (i.e., Target SUM, English-to-target MT, and Target MC). Figure 10 shows the aggregated mean performance across *generative* target language tasks (i.e., Target SUM and English-to-target MT).

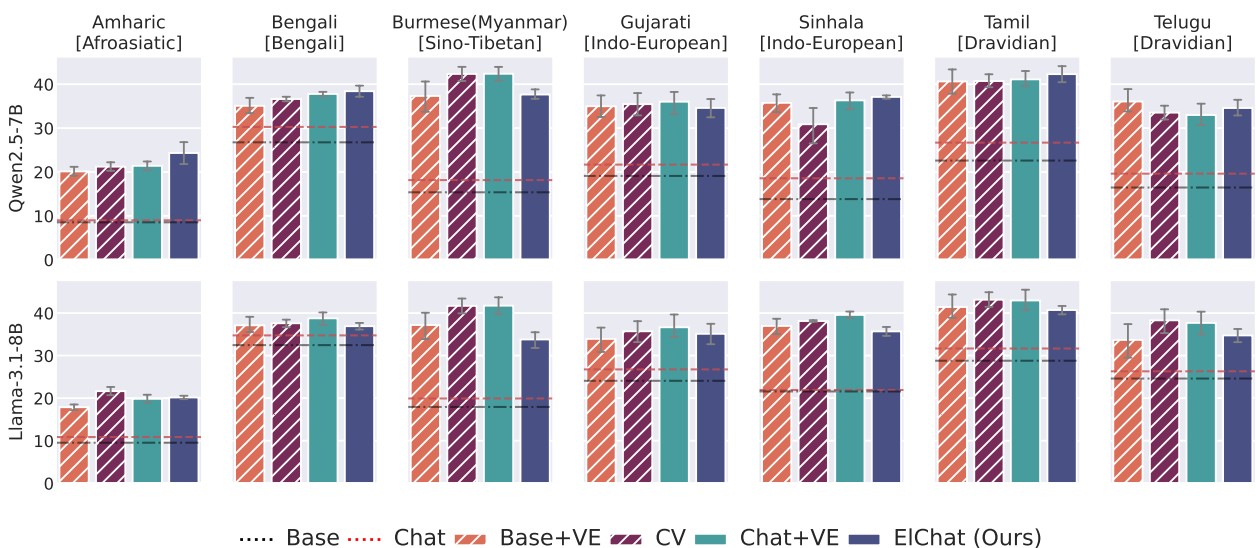

Figure 10: Aggregated mean performance across target language generative tasks (SUM and MT) for each model by language (error bars indicate 95% confidence interval).

**Performance improvements achieved with VE are evident across languages and models.** Overall, we observe from Figure 9 that ElChat consistently outperforms Base and Chat in all the target languages across models.

**The extent to which adapted chat models improve target language task performance substantially varies by language and model.** We observe from Figures 9 and 10 that ElChat substantially improves their target language performance over Base and Chat across models in Amharic, Burmese (Myanmar), and Sinhala. However, its improvement in Bengali remains minimal for both Qwen2.5 and Llama 3.1, especially in generative tasks (Figure 10). Similarly, for Llama 3.1, the performance gains in Tamil, Telugu, and Gujarati are less pronounced, particularly in generative tasks (Figure 10). *We hypothesize that this disparity strongly correlates with the amount of language-specific data used to train the source chat model.* Note that similar findings have already been reported by Yamaguchi et al. (2024a) and Tejaswi et al. (2024), *inter alia.*

**How many languages do source chat LLMs support?** The exact language coverage of the source LLMs remains unclear, as none explicitly list their supported languages.

Qwen2.5 (Qwen et al., 2025; Yang et al., 2024) reportedly supports over 29 languages, including Chinese, English, French, Spanish, Portuguese, German, Italian, Russian, Japanese, Korean, Vietnamese, Thai, and Arabic. However, none of our target languages are explicitly included in this list.

Llama 3.1 (Dubey et al., 2024) officially supports English, German, French, Italian, Portuguese, Hindi, Spanish, and Thai. However, it employs a FastText-based language identification model to categorize documents into 176 languages during pre-processing, suggesting that some of our target languages may be included in its pre-training data.

Additionally, Llama 3.1 reportedly utilizes 15T multilingual tokens (Dubey et al., 2024) for pre-training, while Qwen2.5 uses 18T tokens *in total* (not exclusively multilingual) (Qwen et al., 2025; Yang et al., 2024). This suggests that Llama 3.1 likely benefits from exposure to a broader set of languages. The relatively modest performance improvements observed for Gujarati, Tamil, and Telugu in Llama 3.1 could indicate that these languages were already present in its training data.

**Approximating the language coverage.** We can roughly estimate the language coverage of the source LLMs using MADLAD-400. The rationale is that these open-weight models are generally trained mainly on

a mixture of publicly available data (Gemma Team et al., 2024; Dubey et al., 2024). Given that MADLAD-400 is sourced from CommonCrawl as mentioned in §4, its data distribution can potentially approximate the relative coverage of our target languages.

Table 20 presents the data statistics of MADLAD-400 for our target languages. We observe that Burmese, Amharic, and Sinhala have the fewest articles and total characters in MADLAD-400. Notably, these languages also exhibit the largest performance gains in our experiments. This suggests a strong correlation between the size of language-specific data in MADLAD-400 and the effectiveness of VE in task performance. On the basis of these results, we hypothesize that the two source LLMs used in our experiments might have been trained on very limited language-specific data, or possibly not at all, for Burmese, Amharic, and Sinhala. In contrast, Tamil, Telugu, Bengali, and Gujarati each have over 1 million articles, making them at least 7.4 times larger than the Burmese dataset. This further suggests a higher likelihood of their inclusion in the pre-training data of the source LLMs.

Table 20: Data size of MADLAD-400 Kudugunta et al. (2023) for each language.

| Language | Number of articles | Number of characters |
|----------|-------------------|---------------------|
| Tamil | 5.6M | 10.6B |
| Telugu | 2.5M | 5.1B |
| Bengali | 4.3M | 4.3B |
| Gujarati | 1.3M | 2.1B |
| Sinhala | 788K | 1.9B |
| Amharic | 245.2K | 509M |
| Burmese | 176.5K | 1.3B |

### C.3 Additional Analysis with Newer and Larger Model

We examine the efficacy of ElChat against a newer and larger model compared to Qwen2.5 7B and Llama 3.1 8B. To this end, we employ a state-of-the-art Qwen3 14B model, applying ElChat for adaptation. Due to computational limitations, our evaluation focuses on Amharic, Bengali, and Telugu. Amharic is selected because it shows the most significant speedup gains in target language generative tasks with Qwen2.5 and Llama 3.1 (Tables 17 and 18). Bengali and Telugu are chosen as they are the only languages covered by MGSM. Our analysis will focus on two key questions: (i) *Does Qwen3 exhibit a similar trend in performance with ElChat?* (ii) *If not, what are the potential reasons for the divergence?*

**Safety.** Table 8 shows that the results of Qwen3 align with those of Qwen2.5 in Figure 3. Specifically, CV outperforms ElChat on ToxiGen and ImplicitHate by up to 5 points, while ElChat performs slightly better on TruthfulQA (except for Telugu). Notably, both ElChat and CV generally surpass the Chat baseline across tasks, indicating that they improve safety performance even in this newer, larger model.

**Chat and Instruction-following.** Table 8 demonstrates that ElChat significantly better recovers chat and instruction-following abilities compared to CV across all tasks and languages. For example, ElChat shows drops of up to 6, 6, and 0.41 points from the Chat baseline in IFEval, GSM8K, and MT-Bench, respectively. In contrast, CV experiences considerably larger drops of up to 41, 56, and 0.66 points for the same tasks. A similar trend is observed in AlpacaEval (Table 10). On MGSM (Table 9), both ElChat and CV outperform the Chat baseline; they are competitive in Bengali with a 1-point difference, though CV outperforms ElChat in Telugu by 6 points. Overall, these trends are consistent with our previous observation in §5.1 that ElChat is more effective than CV in enhancing both chat and instruction-following capabilities.

**Target Language.** Contrary to our observations with Qwen2.5 and Llama 3.1 (§5.2; Figure 4, left), ElChat does not consistently outperform Chat in Qwen3 for target language tasks, as shown in Table 13. While we see consistent gains for Amharic, Bengali and Telugu show performance drops in MT, MC, and GMMLU with decreases up to 18 points (e.g., Bengali GMMLU). Similar drops are also observed in Chat+VE and CV, whereas Chat+CPT generally maintains performance comparable to Chat.

We hypothesize that Bengali and Telugu may already be well-represented within Qwen3 due to its significantly enhanced multilingual capabilities. Indeed, Qwen3 was pre-trained on 36T tokens, covering up to 119 languages and dialects. This represents a substantial increase from Qwen2.5, specifically, 29 supported languages and 18T tokens, suggesting a much broader and deeper understanding of various languages. This hypothesis is further supported by Table 36 in Yang et al. (2025), which lists both Bengali and Telugu as officially supported by Qwen3, while Amharic is not.

Finally, when comparing ElChat and CV, we observe they perform competitively across tasks and languages, with a maximum difference of 5.3 points in Amharic MT. This generally aligns with our observations in Qwen2.5 and Llama 3.1.

**Source Language (English).**  Consistent with observations in §5.3 for Qwen2.5 and Llama 3.1, we observe from Table 13 that ElChat generally improves source language (English) performance compared to Chat+VE across tasks and languages. However, these improvements are marginal or modest, with a maximum gain of 6.3 points (Telugu MT). This smaller improvement in English indicates less performance degradation during the VE-process (i.e., less catastrophic forgetting of general English capabilities), suggesting enhanced robustness in Qwen3.

Comparing ElChat with CV, ElChat consistently demonstrates superior or equivalent performance in source language tasks across all languages. This aligns with our observation in §5.3. Notably, the gains tend to be more pronounced in the two generative tasks, SUM and MT, with improvements of up to 19.3 points (Bengali MT).

**Inference Efficiency.**  Finally, Table 19 shows that both ElChat and CV consistently provide speedup gains of up to 3.8x (Amharic SUM with ElChat) in generative tasks within Qwen3. While CV exhibits some reductions in speedup ratios compared to Base+VE (e.g., in Amharic and Bengali), ElChat maintains inference speedups similar to Chat+VE across all tasks and languages. These findings are generally consistent with the results presented in §6 for Qwen2.5 and Llama 3.1.

# D    License

This study uses various publicly available models and datasets with different licenses, as detailed below, all of which permit their use for academic research.

## D.1    Models

Qwen2.5 and Qwen3 are distributed under Apache License 2.0. Llama 3.1 is licensed under the Llama 3 Community License Agreement.[13]

## D.2    Datasets

XL-Sum is licensed under CC BY-NC-SA 4.0. Belebele and FLORES-200 are licensed under CC BY-SA 4.0. BBH, MMLU, GSM8K, and ImplicitHate are distributed under the MIT License. MGSM is distributed under CC BY 4.0. ToxiGen is licensed under Community Data License Agreement - Permissive - Version 2.0. AlpacaEval, IFEval, MT-Bench, and TruthfulQA are distributed under Apache License 2.0.

---

[13]https://llama.meta.com/llama3/license/

Table 3: Prompt template for each task and language.

| Task | Language | Template |
|---|---|---|
| En-Target MT | English | Translate English to {X: a target language}: {sentence} = |
| | Amharic | እንግሊዝኛን ወደ አማርኛ ተርጉም: {sentence} = |
| | Bengali | ইংরেজি থেকে বাংলায় অনুবাদ করুন: {sentence} = |
| | Burmese | အင်္ဂလိပ်မှ မြန်မာသို့ �‌ဘာသာပြန်ပါ။ {sentence} = |
| | Gujarati | અંગ્રેજીમાંથી ગુજરાતીમાં અનુવાદ કરો: {sentence} = |
| | Sinhala | ඉංග්‍රීසි සිංහලයට පරිවර්තනය කරන්න: {sentence} = |
| | Tamil | ஆங்கிலத்திலிருந்து தமிழுக்கு மொழிபெயர்க்கவும்: {sentence} = |
| | Telugu | ఆంగ్లం నుండి తెలుగుకు అనువదించండి: {sentence} = |
| Target-En MT | English | Translate {X: a target language} to English: {sentence} = |
| | Amharic | አማርኛን ወደ እንግሊዝኛ ተርጉም: {sentence} = |
| | Bengali | বাংলা থেকে ইংরেজিতে অনুবাদ করুন: {sentence} = |
| | Burmese | မြန်မာမှ အင်္ဂလိပ်သို့ ဘာသာပြန်ပါ။ {sentence} = |
| | Gujarati | ગુજરાતીમાંથી અંગ્રેજીમાં અનુવાદ કરો: {sentence} = |
| | Sinhala | සිංහලයෙන් ඉංග්‍රීසියට පරිවර්තනය කරන්න: {sentence} = |
| | Tamil | தமிழிலிருந்து ஆங்கிலத்திற்கு மொழிபெயர்க்கவும்: {sentence} = |
| | Telugu | తెలుగు నుండి ఆంగ్లంకు అనువదించండి: {sentence} = |
| SUM | English | Write a short summary of the following text in {language}. Article: {text} Summary: |
| | Amharic | የታችኛው ጽሁፍን በአማርኛ አጭር በማድረግ አሳትረፉት: {text} አጭር መግለጫ: |
| | Bengali | নিম্নলিখিত লেখাটি বাংলায় সংক্ষেপে লিখুন।: {text} সংক্ষিপ্তসার: |
| | Burmese | အောက်ပါစာသားကို မြန်မာဘာသာဖြင့် အကျဉ်းချုပ်ရေးပါ။ ဆောင်းပါး: {text} အကျဉ်းချုပ်: |
| | Gujarati | નીચે આપેલા લખાણને ગુજરાતીમાં સંક્ષિપ્ત લખો.: {text} સંક્ષેપ: |
| | Sinhala | පහත පාඨයේ සාරාංශය සිංහලෙන් ලියන්න. ලිපිය: {text} සාරාංශය: |
| | Tamil | கீழே உள்ள உரையை தமிழில் சுருக்கமாக எழுதுங்கள்: {text} சுருக்கம்: |
| | Telugu | క్రింది వచనం యొక్క సారాంశం తెలుగులో రాయండి. వ్యాసం: {text} సారాంశం: |
| MC | English | {passage} \n Question: {question} \n A. {answer 1}\n B. {answer 2}\n C. {answer 3}\n D. {answer 4}\n Answer: |
| | Amharic | {passage} \n ጥያቄ: {question} \n A. {answer 1}\n B. {answer 2}\n C. {answer 3}\n D. {answer 4}\n መልስ: |
| | Bengali | {passage} \n প্রশ্ন: {question} \n A. {answer 1}\n B. {answer 2}\n C. {answer 3}\n D. {answer 4}\n উত্তর: |
| | Burmese | {passage} \n မေးခွန်း: {question} \n A. {answer 1}\n B. {answer 2}\n C. {answer 3}\n D. {answer 4}\n အဖြေ: |
| | Gujarati | {passage} \n પ્રશ્ન: {question} \n A. {answer 1}\n B. {answer 2}\n C. {answer 3}\n D. {answer 4}\n જવાબ: |
| | Sinhala | {passage} \n ප්‍රශ්නය: {question} \n A. {answer 1}\n B. {answer 2}\n C. {answer 3}\n D. {answer 4}\n පිළිතුර: |
| | Tamil | {passage} \n கேள்வி: {question} \n A. {answer 1}\n B. {answer 2}\n C. {answer 3}\n D. {answer 4}\n பதில்: |
| | Telugu | {passage} \n ప్రశ్న: {question} \n A. {answer 1}\n B. {answer 2}\n C. {answer 3}\n D. {answer 4}\n జవాబు: |

Table 6: Qwen2.5 2 7B chat, instruction-following, and safety performance. Darker blue and red shades indicate higher positive and negative relative performance change over Chat per language and task, respectively. Experiments are limited to models that use the chat template, except for IFEVAL to verify the performance gain of CV over its adapted base model, Base+VE.

| | IFEval | | | | | | | GSM8K | | | | | | | MT-Bench | | | | | | |
|---|---|---|---|---|---|---|---|---|---|---|---|---|---|---|---|---|---|---|---|---|---|
| | am | bn | my | gu | si | ta | te | am | bn | my | gu | si | ta | te | am | bn | my | gu | si | ta | te |
| Base+VE | .19 | .16 | .14 | .15 | .16 | .17 | .14 | - | - | - | - | - | - | - | - | - | - | - | - | - | - |
| CV | .41 | .36 | .45 | .36 | .38 | .38 | .37 | .65 | .66 | .73 | .57 | .56 | .11 | .25 | 2.92 | 3.35 | 3.54 | 3.41 | 3.39 | 3.22 | 3.33 |
| Chat | .70 | .70 | .70 | .70 | .70 | .70 | .70 | .70 | .70 | .70 | .70 | .70 | .70 | .70 | 4.19 | 4.19 | 4.19 | 4.19 | 4.19 | 4.19 | 4.19 |
| Chat+CPT | .40 | .43 | .41 | .31 | .34 | .35 | .36 | .63 | .70 | .66 | .68 | .70 | .60 | .71 | 3.55 | 3.51 | 3.52 | 2.93 | 3.19 | 3.38 | 3.29 |
| Chat+VE | .39 | .38 | .45 | .39 | .40 | .39 | .40 | .43 | .63 | .74 | .47 | .57 | .29 | .44 | 3.12 | 3.43 | 3.67 | 3.31 | 3.50 | 3.34 | 3.21 |
| ElChat | .52 | .54 | .59 | .52 | .53 | .52 | .55 | .66 | .72 | .74 | .68 | .72 | .71 | .70 | 3.30 | 3.59 | 3.81 | 3.58 | 3.75 | 3.69 | 3.58 |

| | TruthfulQA | | | | | | | ToxiGen | | | | | | | ImplicitHate | | | | | | |
|---|---|---|---|---|---|---|---|---|---|---|---|---|---|---|---|---|---|---|---|---|---|
| | am | bn | my | gu | si | ta | te | am | bn | my | gu | si | ta | te | am | bn | my | gu | si | ta | te |
| CV | .55 | .57 | .52 | .58 | .50 | .62 | .60 | .21 | .23 | .22 | .24 | .23 | .22 | .22 | .16 | .20 | .19 | .22 | .20 | .21 | .19 |
| Chat | .31 | .50 | .33 | .41 | .32 | .33 | .32 | .10 | .12 | .07 | .10 | .09 | .09 | .09 | .09 | .12 | .09 | .10 | .07 | .09 | .09 |
| Chat+CPT | .55 | .59 | .38 | .56 | .47 | .59 | .55 | .17 | .17 | .10 | .13 | .13 | .14 | .12 | .16 | .16 | .11 | .13 | .13 | .14 | .12 |
| Chat+VE | .59 | .56 | .51 | .60 | .49 | .59 | .56 | .18 | .21 | .19 | .21 | .21 | .19 | .20 | .14 | .19 | .18 | .20 | .19 | .18 | .18 |
| ElChat | .54 | .60 | .49 | .58 | .53 | .60 | .61 | .12 | .18 | .17 | .17 | .16 | .16 | .16 | .09 | .17 | .17 | .17 | .14 | .15 | .15 |

Table 7: Llama 3.1 8B chat, instruction-following, and safety performance. Darker blue and red shades indicate higher positive and negative relative performance change over Chat per language and task, respectively. (L) stands for linear merging. Experiments are limited to models that use the chat template, except for IFEVAL to verify the performance gain of CV over its adapted base model, Base+VE.

| | IFEval | | | | | | | GSM8K | | | | | | | MT-Bench | | | | | | |
|---|---|---|---|---|---|---|---|---|---|---|---|---|---|---|---|---|---|---|---|---|---|
| | am | bn | my | gu | si | ta | te | am | bn | my | gu | si | ta | te | am | bn | my | gu | si | ta | te |
| Base+VE | .17 | .14 | .13 | .11 | .15 | .12 | .12 | - | - | - | - | - | - | - | - | - | - | - | - | - | - |
| CV | .38 | .39 | .42 | .35 | .41 | .35 | .35 | .10 | .27 | .34 | .30 | .43 | .40 | .48 | 2.93 | 3.10 | 3.49 | 2.84 | 3.03 | 3.04 | 2.88 |
| Chat | .73 | .73 | .73 | .73 | .73 | .73 | .73 | .84 | .84 | .84 | .84 | .84 | .84 | .84 | 3.93 | 3.93 | 3.93 | 3.93 | 3.93 | 3.93 | 3.93 |
| Chat+CPT | .18 | .31 | .37 | .23 | .32 | .36 | .36 | .46 | .51 | .58 | .43 | .51 | .51 | .45 | 2.44 | 2.23 | 2.61 | 2.48 | 2.76 | 2.15 | 2.72 |
| Chat+VE | .33 | .37 | .37 | .32 | .37 | .36 | .28 | .26 | .55 | .38 | .29 | .44 | .35 | .07 | 2.28 | 2.85 | 2.64 | 2.29 | 2.54 | 2.71 | 2.34 |
| ElChat | .45 | .44 | .54 | .44 | .47 | .51 | .47 | .57 | .57 | .72 | .56 | .60 | .56 | .52 | 2.86 | 2.72 | 3.56 | 2.81 | 2.79 | 2.96 | 2.76 |
| ElChat \Merge | .27 | .38 | .41 | .30 | .38 | .33 | .31 | .17 | .54 | .58 | .42 | .55 | .47 | .09 | 2.29 | 2.79 | 2.86 | 2.42 | 2.70 | 2.28 | 2.40 |
| ElChat \Copy | .40 | .37 | .43 | .35 | .40 | .41 | .39 | .15 | .44 | .66 | .14 | .42 | .41 | .36 | 2.61 | 2.82 | 3.09 | 2.45 | 2.41 | 2.70 | 2.48 |
| ElChat (L) | .48 | .44 | .55 | .42 | .50 | .50 | .48 | .56 | .55 | .74 | .55 | .60 | .55 | .51 | 2.91 | 2.97 | 3.58 | 2.69 | 2.76 | 3.23 | 2.75 |

| | TruthfulQA | | | | | | | ToxiGen | | | | | | | ImplicitHate | | | | | | |
|---|---|---|---|---|---|---|---|---|---|---|---|---|---|---|---|---|---|---|---|---|---|
| | am | bn | my | gu | si | ta | te | am | bn | my | gu | si | ta | te | am | bn | my | gu | si | ta | te |
| CV | .11 | .30 | .25 | .10 | .15 | .42 | .39 | .17 | .19 | .19 | .21 | .17 | .18 | .20 | .11 | .17 | .17 | .20 | .15 | .16 | .17 |
| Chat | .34 | .43 | .15 | .46 | .30 | .45 | .40 | .08 | .12 | .07 | .09 | .08 | .09 | .08 | .09 | .11 | .07 | .10 | .08 | .09 | .08 |
| Chat+CPT | .00 | .06 | .80 | .74 | .94 | .00 | .71 | .16 | .17 | .11 | .14 | .13 | .12 | .13 | .18 | .17 | .12 | .15 | .14 | .13 | .13 |
| Chat+VE | .30 | .29 | .09 | .06 | .30 | .27 | .21 | .18 | .22 | .17 | .22 | .19 | .21 | .23 | .14 | .20 | .17 | .21 | .17 | .20 | .21 |
| ElChat | .43 | .44 | .25 | .15 | .48 | .22 | .65 | .10 | .18 | .14 | .18 | .14 | .17 | .15 | .06 | .16 | .14 | .17 | .12 | .16 | .13 |
| ElChat \Merge | .89 | .33 | .09 | .33 | .28 | .77 | .53 | .15 | .21 | .18 | .21 | .19 | .21 | .22 | .06 | .17 | .15 | .18 | .14 | .16 | .14 |
| ElChat \Copy | .42 | .45 | .27 | .16 | .52 | .26 | .60 | .10 | .19 | .15 | .19 | .16 | .18 | .16 | .06 | .17 | .15 | .18 | .14 | .16 | .14 |
| ElChat (L) | .43 | .44 | .23 | .14 | .46 | .24 | .63 | .10 | .18 | .14 | .18 | .14 | .17 | .15 | .06 | .16 | .14 | .17 | .12 | .16 | .13 |

Table 8: Qwen3 14B chat, instruction-following, and safety performance. Darker blue and red shades indicate higher positive and negative relative performance change over Chat per language and task, respectively. Experiments are limited to models that use the chat template, except for IFEval to verify the performance gain of CV over its adapted base model, Base+VE.

|  | IFEval | | | GSM8K | | | MT-Bench | | |
|---|---|---|---|---|---|---|---|---|---|
|  | am | bn | te | am | bn | te | am | bn | te |
| Base+VE | .27 | .27 | .26 | - | - | - | - | - | - |
| CV | .24 | .33 | .33 | .27 | .69 | .40 | 3.58 | 3.78 | 3.77 |
| Chat | .65 | .65 | .65 | .83 | .83 | .83 | 4.24 | 4.24 | 4.24 |
| Chat+CPT | .57 | .62 | .63 | .80 | .84 | .86 | 4.07 | 4.18 | 4.29 |
| Chat+VE | .56 | .58 | .52 | .74 | .79 | .51 | 3.51 | 4.18 | 4.00 |
| ElChat | .59 | .61 | .59 | .77 | .82 | .84 | 3.83 | 4.13 | 4.11 |

|  | TruthfulQA | | | ToxiGen | | | ImplicitHate | | |
|---|---|---|---|---|---|---|---|---|---|
|  | am | bn | te | am | bn | te | am | bn | te |
| CV | .61 | .81 | .69 | .32 | .29 | .26 | .27 | .26 | .25 |
| Chat | .45 | .75 | .66 | .11 | .13 | .08 | .11 | .12 | .08 |
| Chat+CPT | .61 | .81 | .67 | .16 | .10 | .07 | .15 | .10 | .06 |
| Chat+VE | .65 | .88 | .64 | .29 | .26 | .23 | .24 | .24 | .22 |
| ElChat | .63 | .84 | .61 | .27 | .25 | .21 | .22 | .23 | .21 |

Table 9: MGSM performance in Bengali (bn) and Telugu (te) by model. The experimental settings are the same as GSM8K. (L) stands for linear merging.

(a) Qwen2.5 7B

| Model | EM | |
|---|---|---|
|  | bn | te |
| Chat | .23 | .06 |
| CV | .60 | .27 |
| Chat+CPT | .24 | .14 |
| Chat+VE | .39 | .27 |
| ElChat (Ours) | .46 | .35 |

(b) Llama 3.1 8B

| Model | EM | |
|---|---|---|
|  | bn | te |
| Chat | .30 | .12 |
| CV | .31 | .24 |
| Chat+CPT | .12 | .02 |
| Chat+VE | .26 | .28 |
| ElChat (Ours) | .51 | .41 |
| ElChat \Copy | .28 | .36 |
| ElChat \Merge | .43 | .30 |
| ElChat (L) | .53 | .42 |

(c) Qwen3 14B

| Model | EM | |
|---|---|---|
|  | bn | te |
| Chat | .34 | .17 |
| CV | .47 | .28 |
| Chat+CPT | .30 | .16 |
| Chat+VE | .37 | .20 |
| ElChat (Ours) | .48 | .22 |

Table 10: Win rates over GPT-4 (1106 Preview) on ALPACAEVAL v2.0. Darker red shades indicate larger negative relative performance change over Chat per language and base model, respectively. Experiments are limited to Amharic (am), Bengali (bn), and Telugu (te) due to computational constraints.

|  | Qwen2.5 7B | | | Llama 3.1 8B | | | Qwen3 14B | | |
|---|---|---|---|---|---|---|---|---|---|
|  | am | bn | te | am | bn | te | am | bn | te |
| CV | 2.60 | 1.27 | 2.50 | 2.38 | 2.91 | 2.43 | 6.44 | 4.90 | 6.95 |
| Chat | 30.80 | 30.80 | 30.80 | 34.43 | 34.43 | 34.43 | 30.66 | 30.66 | 30.66 |
| Chat+CPT | 4.67 | 3.07 | 3.75 | 2.16 | 1.68 | 2.53 | 10.08 | 24.22 | 27.09 |
| Chat+VE | 2.09 | 3.65 | 2.08 | 2.33 | 2.10 | 1.62 | 8.42 | 9.12 | 6.74 |
| ElChat | 16.60 | 17.16 | 15.92 | 2.72 | 1.96 | 1.82 | 9.33 | 10.46 | 7.39 |

Table 11: Qwen2.5 7B task performance. Darker blue and red shades indicate higher positive and negative relative performance change over Chat per language and task, respectively. Note that GMMLU does not cover Burmese (my), Gujarati (gu), and Tamil (ta).

| | Target sum | | | | | | | English → Target mt | | | | | | | Target mc | | | | | | |
|---|---|---|---|---|---|---|---|---|---|---|---|---|---|---|---|---|---|---|---|---|---|
| | am | bn | my | gu | si | ta | te | am | bn | my | gu | si | ta | te | am | bn | my | gu | si | ta | te |
| Base | $15.8_{0.1}$ | $29.0_{0.1}$ | $21.7_{0.0}$ | $21.9_{0.0}$ | $23.3_{0.1}$ | $26.5_{0.0}$ | $20.2_{0.0}$ | $4.6_{0.1}$ | $32.2_{0.0}$ | $15.7_{0.2}$ | $22.4_{0.1}$ | $10.6_{0.1}$ | $24.8_{0.2}$ | $18.3_{0.0}$ | .32 | .70 | .42 | .57 | .44 | .55 | .46 |
| Base+CPT | $17.3_{0.0}$ | $29.2_{0.0}$ | $21.7_{0.0}$ | $21.4_{0.0}$ | $23.9_{0.0}$ | $26.6_{0.0}$ | $20.4_{0.1}$ | $31.2_{0.1}$ | $42.5_{0.1}$ | $30.9_{0.0}$ | $37.6_{0.1}$ | $35.5_{0.1}$ | $43.8_{0.2}$ | $38.0_{0.1}$ | .74 | .76 | .67 | .70 | .74 | .69 | .66 |
| Base+VE | $18.0_{0.1}$ | $34.4_{0.0}$ | $36.5_{0.1}$ | $29.5_{0.0}$ | $37.3_{0.1}$ | $34.2_{0.3}$ | $31.4_{0.0}$ | $22.6_{0.1}$ | $38.8_{0.1}$ | $44.2_{0.1}$ | $40.3_{0.1}$ | $38.3_{0.1}$ | $46.3_{0.2}$ | $42.5_{0.2}$ | .76 | .76 | .71 | .75 | .78 | .74 | .70 |
| CV | $19.0_{0.1}$ | $35.6_{0.0}$ | $38.5_{0.0}$ | $29.7_{0.1}$ | $38.5_{0.0}$ | $38.5_{0.0}$ | $31.9_{0.0}$ | $22.5_{0.0}$ | $36.9_{0.1}$ | $45.6_{0.3}$ | $39.2_{0.1}$ | $21.2_{0.0}$ | $40.0_{0.1}$ | $32.0_{0.0}$ | .74 | .75 | .73 | .74 | .79 | .76 | .71 |
| Chat | $14.6_{0.1}$ | $30.4_{0.0}$ | $19.1_{0.0}$ | $22.1_{0.0}$ | $22.8_{0.0}$ | $27.7_{0.0}$ | $19.1_{0.0}$ | $3.4_{0.2}$ | $30.2_{0.0}$ | $17.2_{0.1}$ | $21.2_{0.1}$ | $14.3_{0.1}$ | $25.7_{0.1}$ | $20.1_{0.0}$ | .35 | .69 | .39 | .58 | .47 | .54 | .49 |
| Chat+CPT | $18.4_{0.0}$ | $32.2_{0.0}$ | $22.7_{0.0}$ | $24.1_{0.0}$ | $26.9_{0.0}$ | $29.1_{0.0}$ | $22.6_{0.0}$ | $30.9_{0.0}$ | $42.1_{0.0}$ | $30.3_{0.0}$ | $36.5_{0.0}$ | $33.5_{0.0}$ | $42.6_{0.0}$ | $36.3_{0.0}$ | .75 | .75 | .64 | .68 | .73 | .72 | .67 |
| Chat+VE | $19.1_{0.1}$ | $36.7_{0.1}$ | $38.7_{0.0}$ | $29.7_{0.0}$ | $38.9_{0.1}$ | $38.6_{0.1}$ | $32.2_{0.0}$ | $22.9_{0.1}$ | $38.4_{0.1}$ | $45.2_{0.0}$ | $40.0_{0.2}$ | $31.7_{0.5}$ | $40.4_{0.3}$ | $28.9_{0.2}$ | .74 | .76 | .73 | .75 | .78 | .75 | .71 |
| ElChat | $18.7_{0.1}$ | $35.5_{0.1}$ | $36.5_{0.1}$ | $29.6_{0.1}$ | $37.6_{0.1}$ | $37.8_{0.0}$ | $30.7_{0.0}$ | $28.1_{0.2}$ | $40.8_{0.2}$ | $42.4_{0.2}$ | $38.6_{0.2}$ | $37.3_{0.2}$ | $45.8_{0.1}$ | $38.7_{0.1}$ | .70 | .72 | .67 | .68 | .74 | .70 | .63 |

| | English sum | | | | | | | Target → English mt | | | | | | | English mc | | | | | | |
|---|---|---|---|---|---|---|---|---|---|---|---|---|---|---|---|---|---|---|---|---|---|
| | am | bn | my | gu | si | ta | te | am | bn | my | gu | si | ta | te | am | bn | my | gu | si | ta | te |
| Base | $24.4_{0.0}$ | $24.4_{0.0}$ | $24.4_{0.0}$ | $24.4_{0.0}$ | $24.4_{0.0}$ | $24.4_{0.0}$ | $24.4_{0.0}$ | $22.7_{0.1}$ | $52.3_{0.2}$ | $26.6_{0.1}$ | $48.8_{0.2}$ | $27.0_{0.1}$ | $36.7_{0.1}$ | $41.4_{0.3}$ | .92 | .92 | .92 | .92 | .92 | .92 | .92 |
| Base+CPT | $22.6_{0.1}$ | $20.6_{0.2}$ | $23.4_{0.1}$ | $14.9_{0.1}$ | $23.3_{0.1}$ | $23.6_{0.0}$ | $22.0_{0.1}$ | $45.8_{0.1}$ | $44.2_{0.1}$ | $40.8_{0.1}$ | $46.1_{0.2}$ | $43.8_{0.1}$ | $42.4_{0.1}$ | $46.5_{0.2}$ | .90 | .90 | .90 | .90 | .90 | .91 | .89 |
| Base+VE | $24.3_{0.0}$ | $23.6_{0.0}$ | $23.8_{0.0}$ | $23.4_{0.0}$ | $23.5_{0.0}$ | $23.4_{0.0}$ | $23.2_{0.1}$ | $35.6_{0.2}$ | $37.6_{0.2}$ | $37.1_{0.1}$ | $38.9_{0.1}$ | $37.7_{0.1}$ | $36.2_{0.3}$ | $37.6_{0.0}$ | .90 | .90 | .90 | .91 | .91 | .92 | .91 |
| CV | $23.6_{0.0}$ | $22.5_{0.0}$ | $22.9_{0.0}$ | $23.2_{0.0}$ | $23.2_{0.1}$ | $22.9_{0.0}$ | $23.0_{0.1}$ | $31.6_{0.1}$ | $33.7_{0.2}$ | $28.3_{0.6}$ | $31.7_{0.1}$ | $34.1_{0.1}$ | $34.0_{0.2}$ | $36.4_{0.1}$ | .91 | .91 | .91 | .92 | .92 | .92 | .92 |
| Chat | $24.2_{0.0}$ | $24.2_{0.0}$ | $24.2_{0.0}$ | $24.2_{0.0}$ | $24.2_{0.0}$ | $24.2_{0.0}$ | $24.2_{0.0}$ | $25.2_{0.0}$ | $52.9_{0.0}$ | $29.8_{0.1}$ | $49.4_{0.1}$ | $31.1_{0.0}$ | $38.6_{0.0}$ | $42.2_{0.1}$ | .92 | .92 | .92 | .92 | .92 | .92 | .92 |
| Chat+CPT | $23.4_{0.0}$ | $22.3_{0.1}$ | $23.0_{0.0}$ | $20.6_{0.1}$ | $23.3_{0.0}$ | $23.0_{0.0}$ | $22.7_{0.0}$ | $47.0_{0.1}$ | $43.5_{0.0}$ | $29.7_{0.1}$ | $47.9_{0.2}$ | $44.8_{0.0}$ | $40.1_{0.1}$ | $47.9_{0.1}$ | .91 | .91 | .91 | .91 | .92 | .91 | .91 |
| Chat+VE | $24.1_{0.1}$ | $22.9_{0.0}$ | $23.4_{0.1}$ | $23.5_{0.0}$ | $23.5_{0.0}$ | $23.3_{0.0}$ | $23.0_{0.0}$ | $35.2_{0.2}$ | $35.5_{0.1}$ | $33.1_{0.1}$ | $38.2_{0.1}$ | $38.2_{0.1}$ | $37.2_{0.0}$ | $38.5_{0.1}$ | .92 | .92 | .91 | .92 | .91 | .92 | .91 |
| ElChat | $24.4_{0.0}$ | $24.1_{0.0}$ | $24.0_{0.0}$ | $24.1_{0.1}$ | $24.3_{0.0}$ | $24.2_{0.0}$ | $23.8_{0.0}$ | $44.7_{0.0}$ | $44.4_{0.1}$ | $45.4_{0.1}$ | $46.4_{0.0}$ | $45.7_{0.1}$ | $44.8_{0.1}$ | $44.4_{0.1}$ | .91 | .91 | .92 | .91 | .91 | .92 | .91 |

| | Target gmmlu | | | | | | | English mmlu | | | | | | | English bbh | | | | | | |
|---|---|---|---|---|---|---|---|---|---|---|---|---|---|---|---|---|---|---|---|---|---|
| | am | bn | my | gu | si | ta | te | am | bn | my | gu | si | ta | te | am | bn | my | gu | si | ta | te |
| Base | .31 | .47 | - | - | .34 | - | .36 | .74 | .74 | .74 | .74 | .74 | .74 | .74 | .52 | .52 | .52 | .52 | .52 | .52 | .52 |
| Base+CPT | .48 | .53 | - | - | .48 | - | .47 | .71 | .69 | .71 | .71 | .70 | .70 | .70 | .42 | .39 | .43 | .41 | .43 | .43 | .43 |
| Base+VE | .50 | .52 | - | - | .51 | - | .48 | .70 | .71 | .72 | .71 | .71 | .71 | .71 | .41 | .41 | .43 | .42 | .44 | .44 | .42 |
| CV | .49 | .51 | - | - | .51 | - | .47 | .69 | .70 | .69 | .69 | .70 | .69 | .69 | .50 | .48 | .49 | .49 | .47 | .50 | .47 |
| Chat | .31 | .47 | - | - | .35 | - | .35 | .73 | .73 | .73 | .73 | .73 | .73 | .73 | .56 | .56 | .56 | .56 | .56 | .56 | .56 |
| Chat+CPT | .47 | .52 | - | - | .48 | - | .48 | .70 | .71 | .71 | .70 | .70 | .70 | .70 | .51 | .49 | .51 | .49 | .49 | .51 | .50 |
| Chat+VE | .51 | .51 | - | - | .50 | - | .49 | .70 | .70 | .71 | .70 | .70 | .70 | .70 | .48 | .48 | .48 | .48 | .47 | .49 | .49 |
| ElChat | .49 | .46 | - | - | .46 | - | .43 | .71 | .71 | .72 | .71 | .72 | .72 | .71 | .49 | .49 | .51 | .49 | .50 | .51 | .51 |

Table 12: Llama 3.1 8B task performance. Darker blue and red shades indicate higher positive and negative relative performance change over  Chat  per language and task, respectively. (L) stands for linear merging. Note that GMMLU does not cover Burmese (my), Gujarati (gu), and Tamil (ta).

### Target sum / English → Target mt / Target mc

| | Target sum am | bn | my | gu | si | ta | te | E→T mt am | bn | my | gu | si | ta | te | mc am | bn | my | gu | si | ta | te |
|---|---|---|---|---|---|---|---|---|---|---|---|---|---|---|---|---|---|---|---|---|---|
| Base | $11.9_{0.1}$ | $27.5_{0.0}$ | $18.5_{0.0}$ | $19.9_{0.0}$ | $20.2_{0.0}$ | $23.3_{0.1}$ | $18.4_{0.0}$ | $9.8_{0.1}$ | $39.5_{0.0}$ | $21.4_{0.1}$ | $30.0_{0.1}$ | $24.2_{0.1}$ | $36.2_{0.2}$ | $31.0_{0.1}$ | .35 | .62 | .44 | .51 | .52 | .55 | .51 |
| Base+CPT | $10.9_{0.2}$ | $27.5_{0.1}$ | $18.2_{0.0}$ | $19.4_{0.0}$ | $20.8_{0.1}$ | $22.8_{0.0}$ | $18.1_{0.0}$ | $24.3_{0.0}$ | $41.0_{0.0}$ | $26.5_{0.1}$ | $35.9_{0.1}$ | $31.5_{0.1}$ | $39.6_{0.1}$ | $34.5_{0.0}$ | .43 | .46 | .29 | .31 | .34 | .37 | .26 |
| Base+VE | $17.4_{0.0}$ | $34.1_{0.1}$ | $36.7_{0.1}$ | $29.5_{0.0}$ | $36.7_{0.0}$ | $37.4_{0.1}$ | $31.1_{0.0}$ | $19.4_{0.1}$ | $40.7_{0.1}$ | $43.4_{0.2}$ | $39.7_{0.1}$ | $40.3_{0.1}$ | $48.1_{0.1}$ | $42.5_{0.1}$ | .64 | .66 | .58 | .61 | .67 | .57 | .55 |
| CV | $19.5_{0.0}$ | $35.6_{0.0}$ | $37.4_{0.0}$ | $29.8_{0.0}$ | $37.8_{0.0}$ | $38.3_{0.0}$ | $31.4_{0.1}$ | $23.5_{0.1}$ | $38.7_{0.0}$ | $45.1_{0.1}$ | $39.2_{0.1}$ | $38.6_{0.2}$ | $46.2_{0.1}$ | $43.4_{0.0}$ | .53 | .53 | .39 | .44 | .48 | .43 | .48 |
| Chat | $12.2_{0.0}$ | $29.8_{0.0}$ | $18.6_{0.0}$ | $22.4_{0.1}$ | $22.0_{0.0}$ | $26.5_{0.0}$ | $20.4_{0.1}$ | $9.5_{0.1}$ | $39.7_{0.1}$ | $21.3_{0.0}$ | $31.1_{0.0}$ | $21.8_{0.0}$ | $36.8_{0.0}$ | $32.2_{0.1}$ | .36 | .65 | .42 | .56 | .62 | .61 | .59 |
| Chat+CPT | $11.2_{0.0}$ | $26.8_{0.0}$ | $18.8_{0.0}$ | $20.9_{0.1}$ | $20.7_{0.0}$ | $23.4_{0.0}$ | $19.6_{0.1}$ | $24.1_{0.0}$ | $40.0_{0.1}$ | $26.0_{0.1}$ | $34.9_{0.0}$ | $31.3_{0.0}$ | $39.3_{0.1}$ | $34.7_{0.1}$ | .34 | .32 | .28 | .26 | .29 | .31 | .27 |
| Chat+VE | $17.9_{0.2}$ | $35.4_{0.0}$ | $37.1_{0.0}$ | $29.9_{0.0}$ | $38.1_{0.2}$ | $37.6_{0.0}$ | $31.7_{0.1}$ | $22.0_{0.4}$ | $41.2_{0.1}$ | $45.9_{0.1}$ | $41.3_{0.1}$ | $41.3_{0.2}$ | $49.0_{0.1}$ | $43.3_{0.2}$ | .61 | .58 | .43 | .59 | .59 | .50 | .52 |
| ElChat | $19.4_{0.1}$ | $35.1_{0.1}$ | $36.4_{0.1}$ | $29.6_{0.1}$ | $38.1_{0.1}$ | $38.4_{0.0}$ | $31.2_{0.1}$ | $21.1_{0.1}$ | $38.1_{0.1}$ | $36.1_{0.2}$ | $38.9_{0.2}$ | $34.6_{0.2}$ | $42.5_{0.3}$ | $37.4_{0.3}$ | .64 | .67 | .53 | .54 | .67 | .55 | .56 |
| ElChat \Merge | $16.2_{0.2}$ | $33.9_{0.1}$ | $36.7_{0.1}$ | $29.4_{0.0}$ | $37.3_{0.1}$ | $36.7_{0.0}$ | $31.4_{0.1}$ | $22.1_{0.1}$ | $40.7_{0.1}$ | $45.2_{0.2}$ | $41.4_{0.2}$ | $36.9_{0.3}$ | $47.5_{0.0}$ | $43.0_{0.1}$ | .50 | .52 | .42 | .53 | .52 | .48 | .49 |
| ElChat \Copy | $19.3_{0.2}$ | $35.6_{0.1}$ | $36.5_{0.1}$ | $29.6_{0.1}$ | $38.6_{0.1}$ | $38.9_{0.2}$ | $31.4_{0.1}$ | $19.9_{0.1}$ | $37.9_{0.1}$ | $35.5_{0.2}$ | $38.9_{0.2}$ | $35.1_{0.1}$ | $42.9_{0.2}$ | $38.1_{0.1}$ | .67 | .69 | .57 | .57 | .72 | .59 | .61 |
| ElChat (L) | $19.3_{0.2}$ | $35.3_{0.0}$ | $36.1_{0.0}$ | $29.3_{0.1}$ | $38.3_{0.0}$ | $38.4_{0.1}$ | $31.1_{0.1}$ | $20.8_{0.1}$ | $37.7_{0.1}$ | $35.2_{0.3}$ | $38.4_{0.2}$ | $33.8_{0.2}$ | $41.5_{0.4}$ | $36.6_{0.3}$ | .64 | .66 | .53 | .53 | .66 | .56 | .56 |

### English sum / Target → English mt / English mc

| | Eng sum am | bn | my | gu | si | ta | te | T→E mt am | bn | my | gu | si | ta | te | mc am | bn | my | gu | si | ta | te |
|---|---|---|---|---|---|---|---|---|---|---|---|---|---|---|---|---|---|---|---|---|---|
| Base | $23.5_{0.0}$ | $23.5_{0.0}$ | $23.5_{0.0}$ | $23.5_{0.0}$ | $23.5_{0.0}$ | $23.5_{0.0}$ | $23.5_{0.0}$ | $33.3_{0.1}$ | $58.6_{0.1}$ | $44.4_{0.2}$ | $57.9_{0.2}$ | $49.9_{0.2}$ | $53.1_{0.2}$ | $56.3_{0.2}$ | .88 | .88 | .88 | .88 | .88 | .88 | .88 |
| Base+CPT | $19.0_{0.1}$ | $17.2_{0.1}$ | $21.7_{0.2}$ | $17.0_{0.1}$ | $19.4_{0.0}$ | $22.3_{0.1}$ | $21.8_{0.1}$ | $44.7_{0.2}$ | $47.1_{0.1}$ | $42.9_{0.1}$ | $50.3_{0.1}$ | $46.4_{0.2}$ | $41.5_{0.0}$ | $47.7_{0.2}$ | .78 | .82 | .81 | .83 | .80 | .80 | .81 |
| Base+VE | $22.5_{0.1}$ | $22.6_{0.2}$ | $22.9_{0.0}$ | $22.9_{0.0}$ | $23.0_{0.1}$ | $23.1_{0.0}$ | $23.0_{0.0}$ | $36.0_{0.1}$ | $40.2_{0.1}$ | $37.5_{0.1}$ | $42.6_{0.1}$ | $39.0_{0.1}$ | $37.8_{0.2}$ | $41.0_{0.0}$ | .82 | .82 | .81 | .83 | .84 | .82 | .81 |
| CV | $23.4_{0.0}$ | $23.3_{0.1}$ | $23.6_{0.1}$ | $23.1_{0.0}$ | $23.2_{0.0}$ | $23.5_{0.1}$ | $23.0_{0.0}$ | $33.1_{0.2}$ | $35.6_{0.2}$ | $30.7_{0.1}$ | $37.8_{0.1}$ | $32.4_{0.1}$ | $34.2_{0.1}$ | $35.8_{0.1}$ | .83 | .83 | .85 | .82 | .84 | .80 | .83 |
| Chat | $26.8_{0.0}$ | $26.8_{0.0}$ | $26.8_{0.0}$ | $26.8_{0.0}$ | $26.8_{0.0}$ | $26.8_{0.0}$ | $26.8_{0.0}$ | $29.3_{0.1}$ | $57.3_{0.2}$ | $36.2_{0.2}$ | $57.3_{0.2}$ | $47.9_{0.2}$ | $53.1_{0.1}$ | $56.8_{0.2}$ | .91 | .91 | .91 | .91 | .91 | .91 | .91 |
| Chat+CPT | $22.6_{0.1}$ | $22.7_{0.1}$ | $21.4_{0.2}$ | $22.0_{0.0}$ | $24.1_{0.1}$ | $23.5_{0.0}$ | $23.3_{0.0}$ | $31.2_{0.1}$ | $36.5_{0.3}$ | $37.1_{0.4}$ | $40.6_{0.3}$ | $42.1_{0.1}$ | $39.1_{0.3}$ | $43.9_{0.3}$ | .28 | .83 | .82 | .84 | .83 | .80 | .82 |
| Chat+VE | $23.4_{0.0}$ | $23.5_{0.0}$ | $23.8_{0.0}$ | $23.0_{0.1}$ | $23.7_{0.1}$ | $23.5_{0.1}$ | $23.4_{0.1}$ | $26.5_{0.0}$ | $36.8_{0.1}$ | $33.5_{0.1}$ | $38.0_{0.2}$ | $33.5_{0.2}$ | $36.0_{0.1}$ | $34.9_{0.1}$ | .84 | .86 | .85 | .83 | .85 | .84 | .83 |
| ElChat | $24.1_{0.1}$ | $23.5_{0.1}$ | $24.1_{0.0}$ | $23.6_{0.1}$ | $23.8_{0.0}$ | $24.2_{0.1}$ | $23.7_{0.0}$ | $27.5_{0.1}$ | $27.6_{0.3}$ | $31.7_{0.2}$ | $35.6_{0.0}$ | $25.1_{0.4}$ | $31.5_{0.3}$ | $34.0_{0.2}$ | .90 | .89 | .89 | .90 | .90 | .88 | .90 |
| ElChat \Merge | $23.3_{0.1}$ | $23.1_{0.0}$ | $23.8_{0.1}$ | $22.8_{0.1}$ | $23.5_{0.1}$ | $20.9_{0.1}$ | $23.1_{0.1}$ | $14.4_{0.4}$ | $38.5_{0.0}$ | $34.6_{0.1}$ | $37.2_{0.1}$ | $30.3_{0.0}$ | $35.9_{0.1}$ | $35.6_{0.2}$ | .80 | .82 | .83 | .47 | .79 | .55 | .78 |
| ElChat \Copy | $24.2_{0.0}$ | $24.0_{0.1}$ | $24.1_{0.0}$ | $24.0_{0.0}$ | $24.0_{0.1}$ | $24.3_{0.1}$ | $24.1_{0.1}$ | $27.7_{0.3}$ | $27.0_{0.2}$ | $32.1_{0.3}$ | $36.3_{0.3}$ | $27.8_{0.3}$ | $30.6_{0.1}$ | $31.5_{0.2}$ | .90 | .89 | .89 | .90 | .89 | .89 | .90 |
| ElChat (L) | $24.0_{0.0}$ | $23.5_{0.1}$ | $24.0_{0.0}$ | $23.8_{0.0}$ | $23.8_{0.1}$ | $24.2_{0.0}$ | $23.7_{0.0}$ | $27.9_{0.2}$ | $26.6_{0.1}$ | $31.6_{0.2}$ | $36.1_{0.4}$ | $24.1_{0.3}$ | $31.5_{0.1}$ | $33.6_{0.2}$ | .90 | .89 | .89 | .90 | .90 | .88 | .91 |

### Target gmmlu / English mmlu / English bbh

| | gmmlu am | bn | my | gu | si | ta | te | mmlu am | bn | my | gu | si | ta | te | bbh am | bn | my | gu | si | ta | te |
|---|---|---|---|---|---|---|---|---|---|---|---|---|---|---|---|---|---|---|---|---|---|
| Base | .31 | .41 | - | - | .36 | - | .39 | .65 | .65 | .65 | .65 | .65 | .65 | .65 | .46 | .46 | .46 | .46 | .46 | .46 | .46 |
| Base+CPT | .32 | .34 | - | - | .32 | - | .32 | .49 | .55 | .57 | .54 | .54 | .50 | .54 | .36 | .36 | .41 | .37 | .38 | .36 | .37 |
| Base+VE | .39 | .42 | - | - | .42 | - | .38 | .55 | .56 | .58 | .53 | .57 | .55 | .53 | .37 | .39 | .38 | .37 | .38 | .36 | .38 |
| CV | .32 | .31 | - | - | .34 | - | .33 | .48 | .50 | .53 | .47 | .51 | .48 | .49 | .41 | .40 | .40 | .40 | .40 | .38 | .41 |
| Chat | .29 | .42 | - | - | .36 | - | .39 | .67 | .67 | .67 | .67 | .67 | .67 | .67 | .56 | .56 | .56 | .56 | .56 | .56 | .56 |
| Chat+CPT | .28 | .26 | - | - | .26 | - | .29 | .27 | .53 | .54 | .54 | .52 | .47 | .53 | .37 | .39 | .40 | .39 | .39 | .38 | .40 |
| Chat+VE | .36 | .33 | - | - | .35 | - | .36 | .51 | .51 | .55 | .52 | .54 | .54 | .52 | .40 | .39 | .41 | .40 | .40 | .38 | .40 |
| ElChat | .37 | .36 | - | - | .38 | - | .36 | .61 | .57 | .60 | .58 | .61 | .60 | .60 | .47 | .43 | .47 | .43 | .44 | .42 | .45 |
| ElChat \Merge | .35 | .32 | - | - | .33 | - | .35 | .44 | .48 | .53 | .32 | .50 | .34 | .49 | .37 | .37 | .40 | .37 | .38 | .36 | .39 |
| ElChat \Copy | .37 | .37 | - | - | .39 | - | .37 | .61 | .57 | .61 | .58 | .61 | .60 | .60 | .47 | .44 | .47 | .44 | .43 | .43 | .45 |
| ElChat (L) | .37 | .36 | - | - | .38 | - | .36 | .61 | .57 | .61 | .58 | .61 | .59 | .60 | .47 | .43 | .47 | .43 | .44 | .43 | .46 |

Table 13: Qwen3 14B task performance. Darker blue and red shades indicate higher positive and negative relative performance change over Chat per language and task, respectively.

| | Target sum | | | English → Target mt | | | Target mc | | |
|---|---|---|---|---|---|---|---|---|---|
| | am | bn | te | am | bn | te | am | bn | te |
| Base | $17.0_{0.0}$ | $28.7_{0.2}$ | $20.7_{0.1}$ | $13.5_{0.3}$ | $42.2_{0.1}$ | $36.4_{0.1}$ | .56 | .84 | .81 |
| Base+CPT | $17.0_{0.0}$ | $31.1_{0.0}$ | $20.9_{0.0}$ | $28.4_{0.1}$ | $43.5_{0.0}$ | $38.7_{0.1}$ | .77 | .85 | .79 |
| Base+VE | $17.0_{0.1}$ | $34.0_{0.1}$ | $30.5_{0.1}$ | $19.7_{0.1}$ | $32.9_{0.2}$ | $33.7_{0.2}$ | .78 | .80 | .72 |
| CV | $19.5_{0.1}$ | $35.7_{0.1}$ | $31.2_{0.0}$ | $22.4_{0.1}$ | $33.4_{0.1}$ | $33.5_{0.1}$ | .75 | .72 | .63 |
| Chat | $17.0_{0.0}$ | $32.0_{0.0}$ | $22.7_{0.0}$ | $13.6_{0.1}$ | $41.3_{0.0}$ | $33.9_{0.0}$ | .53 | .84 | .76 |
| Chat+CPT | $19.1_{0.0}$ | $32.1_{0.0}$ | $21.8_{0.0}$ | $26.2_{0.1}$ | $41.8_{0.1}$ | $36.8_{0.1}$ | .76 | .83 | .76 |
| Chat+VE | $19.5_{0.0}$ | $35.3_{0.1}$ | $30.4_{0.0}$ | $23.7_{0.1}$ | $33.9_{0.0}$ | $32.1_{0.2}$ | .78 | .74 | .65 |
| ElChat | $19.6_{0.1}$ | $35.5_{0.0}$ | $30.4_{0.0}$ | $27.7_{0.3}$ | $34.9_{0.2}$ | $31.7_{0.0}$ | .78 | .75 | .64 |

| | English sum | | | Target → English mt | | | English mc | | |
|---|---|---|---|---|---|---|---|---|---|
| | am | bn | te | am | bn | te | am | bn | te |
| Base | $24.5_{0.0}$ | $24.5_{0.0}$ | $24.5_{0.0}$ | $39.2_{0.1}$ | $60.5_{0.0}$ | $60.7_{0.2}$ | .94 | .94 | .94 |
| Base+CPT | $23.1_{0.0}$ | $23.5_{0.0}$ | $23.3_{0.0}$ | $47.9_{0.1}$ | $49.7_{0.0}$ | $51.4_{0.3}$ | .94 | .94 | .94 |
| Base+VE | $24.2_{0.0}$ | $23.1_{0.0}$ | $23.8_{0.1}$ | $39.6_{0.3}$ | $38.5_{0.1}$ | $35.3_{0.2}$ | .95 | .95 | .94 |
| CV | $22.9_{0.0}$ | $22.6_{0.0}$ | $23.0_{0.0}$ | $42.7_{0.2}$ | $31.8_{0.1}$ | $35.9_{0.1}$ | .94 | .93 | .94 |
| Chat | $24.2_{0.0}$ | $24.2_{0.0}$ | $24.2_{0.0}$ | $38.6_{0.0}$ | $59.6_{0.0}$ | $59.6_{0.0}$ | .94 | .94 | .94 |
| Chat+CPT | $23.8_{0.0}$ | $24.2_{0.0}$ | $24.1_{0.0}$ | $54.8_{0.1}$ | $59.0_{0.0}$ | $59.1_{0.1}$ | .93 | .93 | .93 |
| Chat+VE | $24.2_{0.0}$ | $23.6_{0.0}$ | $24.1_{0.0}$ | $51.0_{0.2}$ | $50.6_{0.0}$ | $41.6_{0.2}$ | .93 | .94 | .94 |
| ElChat | $24.1_{0.0}$ | $23.9_{0.0}$ | $24.4_{0.0}$ | $53.0_{0.1}$ | $51.1_{0.1}$ | $47.3_{0.1}$ | .94 | .94 | .94 |

| | Target gmmlu | | | English mmlu | | | English bbh | | |
|---|---|---|---|---|---|---|---|---|---|
| | am | bn | te | am | bn | te | am | bn | te |
| Base | .42 | .64 | .61 | .80 | .80 | .80 | .63 | .63 | .63 |
| Base+CPT | .54 | .64 | .60 | .80 | .80 | .80 | .59 | .58 | .58 |
| Base+VE | .57 | .49 | .45 | .79 | .80 | .80 | .55 | .56 | .56 |
| CV | .51 | .39 | .37 | .77 | .77 | .76 | .62 | .63 | .62 |
| Chat | .37 | .58 | .55 | .77 | .77 | .77 | .63 | .63 | .63 |
| Chat+CPT | .49 | .57 | .56 | .76 | .54 | .71 | .65 | .65 | .65 |
| Chat+VE | .52 | .39 | .38 | .76 | .76 | .76 | .64 | .63 | .61 |
| ElChat | .52 | .40 | .38 | .77 | .77 | .77 | .64 | .64 | .62 |

Table 14: Qwen2.5 7B task performance. Darker blue and red shades indicate higher positive and negative relative performance change over Chat per language and task, respectively.

| | Target sum [ROUGE-L] | | | | | | | English → Target mt [chrF++] | | | | | | |
|---|---|---|---|---|---|---|---|---|---|---|---|---|---|---|
| | am | bn | my | gu | si | ta | te | am | bn | my | gu | si | ta | te |
| Base | $1.8_{0.1}$ | $1.7_{0.0}$ | $7.2_{0.0}$ | $2.8_{0.1}$ | $42.4_{0.5}$ | $2.4_{0.0}$ | $18.7_{0.1}$ | $3.8_{0.1}$ | $28.3_{0.1}$ | $12.3_{0.2}$ | $19.6_{0.1}$ | $9.9_{0.0}$ | $20.7_{0.1}$ | $15.3_{0.1}$ |
| Base+CPT | $2.1_{0.1}$ | $1.8_{0.0}$ | $5.5_{0.0}$ | $3.0_{0.0}$ | $42.5_{0.2}$ | $2.2_{0.0}$ | $19.0_{0.2}$ | $28.4_{0.1}$ | $38.0_{0.1}$ | $24.5_{0.0}$ | $34.0_{0.1}$ | $32.8_{0.1}$ | $38.1_{0.2}$ | $33.7_{0.1}$ |
| Base+VE | $3.0_{0.3}$ | $1.7_{0.1}$ | $9.2_{0.2}$ | $4.2_{0.2}$ | $55.5_{0.2}$ | $2.8_{0.1}$ | $24.9_{0.3}$ | $20.5_{0.1}$ | $34.6_{0.1}$ | $35.0_{0.1}$ | $36.8_{0.1}$ | $35.7_{0.1}$ | $40.1_{0.2}$ | $37.8_{0.1}$ |
| CV | $4.0_{0.1}$ | $1.8_{0.0}$ | $9.7_{0.0}$ | $5.2_{0.1}$ | $55.1_{0.2}$ | $4.1_{0.0}$ | $23.7_{0.1}$ | $20.1_{0.0}$ | $32.8_{0.1}$ | $36.3_{0.2}$ | $35.8_{0.1}$ | $20.2_{0.0}$ | $34.6_{0.1}$ | $28.2_{0.1}$ |
| Chat | $1.5_{0.1}$ | $1.3_{0.0}$ | $3.9_{0.1}$ | $3.0_{0.1}$ | $35.9_{0.1}$ | $2.2_{0.0}$ | $15.0_{0.1}$ | $2.9_{0.1}$ | $26.6_{0.0}$ | $13.4_{0.0}$ | $18.8_{0.1}$ | $13.9_{0.1}$ | $21.5_{0.1}$ | $16.9_{0.0}$ |
| Chat+CPT | $2.5_{0.1}$ | $1.4_{0.0}$ | $4.6_{0.2}$ | $5.2_{0.0}$ | $49.4_{0.1}$ | $2.4_{0.0}$ | $20.7_{0.2}$ | $28.2_{0.1}$ | $37.8_{0.0}$ | $24.2_{0.0}$ | $33.0_{0.0}$ | $31.0_{0.1}$ | $36.9_{0.0}$ | $32.3_{0.0}$ |
| Chat+VE | $3.7_{0.4}$ | $1.6_{0.0}$ | $9.8_{0.1}$ | $4.8_{0.1}$ | $56.0_{0.4}$ | $3.6_{0.1}$ | $25.1_{0.2}$ | $20.6_{0.1}$ | $34.2_{0.1}$ | $35.9_{0.1}$ | $36.6_{0.2}$ | $29.9_{0.5}$ | $34.9_{0.3}$ | $25.4_{0.2}$ |
| ElChat | $4.0_{0.2}$ | $2.0_{0.0}$ | $8.5_{0.3}$ | $4.1_{0.1}$ | $56.1_{0.4}$ | $3.9_{0.2}$ | $24.1_{0.2}$ | $25.5_{0.2}$ | $36.3_{0.2}$ | $33.5_{0.1}$ | $35.7_{0.1}$ | $34.8_{0.2}$ | $39.6_{0.1}$ | $34.3_{0.1}$ |
| | English sum [ROUGE-L] | | | | | | | Target → English mt [chrF++] | | | | | | |
| | am | bn | my | gu | si | ta | te | am | bn | my | gu | si | ta | te |
| Base | $12.0_{0.0}$ | $12.0_{0.0}$ | $12.0_{0.0}$ | $12.0_{0.0}$ | $12.0_{0.0}$ | $12.0_{0.0}$ | $12.0_{0.0}$ | $20.6_{0.1}$ | $50.2_{0.2}$ | $24.7_{0.2}$ | $46.6_{0.1}$ | $25.1_{0.0}$ | $34.8_{0.1}$ | $39.2_{0.3}$ |
| Base+CPT | $11.0_{0.1}$ | $10.1_{0.1}$ | $11.1_{0.0}$ | $7.2_{0.0}$ | $10.9_{0.0}$ | $11.1_{0.0}$ | $10.2_{0.1}$ | $43.0_{0.1}$ | $41.4_{0.1}$ | $38.6_{0.1}$ | $43.4_{0.1}$ | $40.6_{0.1}$ | $40.1_{0.1}$ | $43.8_{0.2}$ |
| Base+VE | $12.0_{0.1}$ | $11.4_{0.1}$ | $11.7_{0.1}$ | $11.0_{0.0}$ | $11.0_{0.0}$ | $10.9_{0.1}$ | $11.1_{0.1}$ | $33.4_{0.1}$ | $35.4_{0.2}$ | $36.1_{0.1}$ | $36.5_{0.1}$ | $35.0_{0.1}$ | $34.9_{0.3}$ | $35.7_{0.1}$ |
| CV | $11.5_{0.0}$ | $10.8_{0.0}$ | $10.9_{0.0}$ | $11.2_{0.0}$ | $11.0_{0.0}$ | $10.8_{0.0}$ | $10.9_{0.0}$ | $29.7_{0.1}$ | $31.9_{0.2}$ | $30.1_{0.2}$ | $30.0_{0.1}$ | $32.2_{0.1}$ | $32.3_{0.1}$ | $34.1_{0.1}$ |
| Chat | $11.8_{0.0}$ | $11.8_{0.0}$ | $11.8_{0.0}$ | $11.8_{0.0}$ | $11.8_{0.0}$ | $11.8_{0.0}$ | $11.8_{0.0}$ | $22.8_{0.0}$ | $50.3_{0.0}$ | $27.5_{0.0}$ | $46.9_{0.1}$ | $28.6_{0.0}$ | $36.2_{0.0}$ | $39.6_{0.0}$ |
| Chat+CPT | $11.5_{0.1}$ | $10.7_{0.0}$ | $10.9_{0.0}$ | $9.9_{0.1}$ | $11.1_{0.0}$ | $11.1_{0.0}$ | $10.8_{0.0}$ | $44.5_{0.0}$ | $40.7_{0.1}$ | $28.2_{0.1}$ | $45.4_{0.2}$ | $41.8_{0.0}$ | $38.1_{0.1}$ | $45.5_{0.1}$ |
| Chat+VE | $11.9_{0.0}$ | $11.0_{0.0}$ | $11.0_{0.0}$ | $11.2_{0.0}$ | $11.2_{0.0}$ | $11.1_{0.0}$ | $10.9_{0.0}$ | $33.0_{0.2}$ | $34.4_{0.0}$ | $31.4_{0.1}$ | $36.3_{0.0}$ | $36.3_{0.1}$ | $35.4_{0.1}$ | $36.0_{0.1}$ |
| ElChat | $12.2_{0.0}$ | $11.5_{0.0}$ | $11.3_{0.0}$ | $11.6_{0.0}$ | $11.7_{0.0}$ | $11.6_{0.0}$ | $11.4_{0.0}$ | $42.9_{0.0}$ | $42.3_{0.1}$ | $42.8_{0.1}$ | $44.4_{0.0}$ | $43.4_{0.1}$ | $42.5_{0.1}$ | $42.2_{0.1}$ |

Table 15: Llama 3.1 8B task performance. Darker blue and red shades indicate higher positive and negative relative performance change over Chat per language and task, respectively. (L) stands for linear merging.

| | Target sum [ROUGE-L] | | | | | | | English → Target mt [chrF++] | | | | | | |
|---|---|---|---|---|---|---|---|---|---|---|---|---|---|---|
| | am | bn | my | gu | si | ta | te | am | bn | my | gu | si | ta | te |
| Base | $1.6_{0.1}$ | $1.9_{0.0}$ | $6.2_{0.1}$ | $3.2_{0.0}$ | $36.3_{0.4}$ | $2.0_{0.1}$ | $18.3_{0.2}$ | $8.4_{0.1}$ | $35.6_{0.0}$ | $16.9_{0.1}$ | $27.0_{0.1}$ | $22.5_{0.1}$ | $31.5_{0.2}$ | $27.3_{0.1}$ |
| Base+CPT | $1.0_{0.1}$ | $1.9_{0.0}$ | $6.3_{0.0}$ | $2.9_{0.2}$ | $38.2_{0.2}$ | $1.6_{0.1}$ | $17.3_{0.2}$ | $22.0_{0.0}$ | $36.8_{0.0}$ | $21.1_{0.1}$ | $32.5_{0.1}$ | $29.0_{0.1}$ | $34.5_{0.1}$ | $30.5_{0.0}$ |
| Base+VE | $3.2_{0.2}$ | $1.6_{0.0}$ | $8.8_{0.4}$ | $4.6_{0.1}$ | $55.0_{0.2}$ | $3.3_{0.0}$ | $25.2_{0.1}$ | $17.5_{0.1}$ | $36.4_{0.1}$ | $34.3_{0.1}$ | $36.4_{0.1}$ | $37.4_{0.1}$ | $41.9_{0.1}$ | $37.7_{0.1}$ |
| CV | $3.1_{0.2}$ | $1.7_{0.0}$ | $8.2_{0.1}$ | $5.1_{0.2}$ | $55.5_{0.4}$ | $4.1_{0.0}$ | $25.8_{0.3}$ | $21.4_{0.1}$ | $34.0_{0.1}$ | $35.6_{0.1}$ | $32.5_{0.2}$ | $35.6_{0.1}$ | $40.3_{0.1}$ | $38.6_{0.0}$ |
| Chat | $1.7_{0.0}$ | $1.1_{0.0}$ | $4.2_{0.0}$ | $3.1_{0.0}$ | $46.5_{0.0}$ | $3.0_{0.0}$ | $21.3_{0.1}$ | $8.0_{0.0}$ | $35.7_{0.1}$ | $16.8_{0.0}$ | $28.1_{0.0}$ | $20.9_{0.1}$ | $32.1_{0.0}$ | $28.7_{0.1}$ |
| Chat+CPT | $1.6_{0.0}$ | $1.7_{0.0}$ | $3.5_{0.3}$ | $2.8_{0.1}$ | $36.5_{0.4}$ | $2.1_{0.0}$ | $18.2_{0.2}$ | $21.9_{0.1}$ | $35.7_{0.1}$ | $20.6_{0.1}$ | $31.5_{0.0}$ | $28.9_{0.1}$ | $34.3_{0.1}$ | $30.7_{0.1}$ |
| Chat+VE | $2.7_{0.1}$ | $1.4_{0.1}$ | $8.8_{0.1}$ | $4.8_{0.1}$ | $54.7_{0.3}$ | $3.4_{0.1}$ | $25.9_{0.3}$ | $20.1_{0.3}$ | $37.0_{0.1}$ | $36.5_{0.1}$ | $38.0_{0.2}$ | $38.6_{0.1}$ | $42.8_{0.1}$ | $38.8_{0.2}$ |
| ElChat | $3.4_{0.3}$ | $1.3_{0.0}$ | $7.5_{0.3}$ | $4.8_{0.1}$ | $53.9_{0.2}$ | $4.3_{0.3}$ | $25.4_{0.3}$ | $19.1_{0.1}$ | $33.8_{0.1}$ | $28.4_{0.1}$ | $35.6_{0.2}$ | $31.9_{0.2}$ | $36.4_{0.3}$ | $33.0_{0.2}$ |
| ElChat \Merge | $2.5_{0.3}$ | $1.6_{0.1}$ | $8.9_{0.2}$ | $5.0_{0.1}$ | $54.2_{0.3}$ | $2.9_{0.1}$ | $25.8_{0.2}$ | $20.4_{0.1}$ | $36.5_{0.1}$ | $36.0_{0.0}$ | $38.6_{0.2}$ | $36.2_{0.2}$ | $41.6_{0.0}$ | $38.6_{0.1}$ |
| ElChat \Copy | $3.2_{0.1}$ | $1.4_{0.0}$ | $7.6_{0.3}$ | $5.0_{0.0}$ | $53.7_{0.1}$ | $4.4_{0.2}$ | $25.5_{0.2}$ | $18.1_{0.1}$ | $33.6_{0.1}$ | $28.0_{0.2}$ | $35.8_{0.2}$ | $32.6_{0.1}$ | $37.1_{0.2}$ | $33.6_{0.1}$ |
| ElChat (L) | $3.7_{0.1}$ | $1.3_{0.0}$ | $7.9_{0.2}$ | $5.0_{0.1}$ | $54.1_{0.5}$ | $4.5_{0.1}$ | $25.8_{0.2}$ | $18.9_{0.1}$ | $33.4_{0.1}$ | $27.8_{0.3}$ | $35.0_{0.2}$ | $31.1_{0.1}$ | $35.6_{0.4}$ | $32.2_{0.3}$ |
| | English sum [ROUGE-L] | | | | | | | Target → English mt [chrF++] | | | | | | |
| | am | bn | my | gu | si | ta | te | am | bn | my | gu | si | ta | te |
| Base | $10.9_{0.0}$ | $10.9_{0.0}$ | $10.9_{0.0}$ | $10.9_{0.0}$ | $10.9_{0.0}$ | $10.9_{0.0}$ | $10.9_{0.0}$ | $31.4_{0.1}$ | $56.3_{0.1}$ | $42.2_{0.2}$ | $55.8_{0.2}$ | $47.8_{0.2}$ | $50.9_{0.1}$ | $54.1_{0.2}$ |
| Base+CPT | $8.7_{0.1}$ | $7.8_{0.0}$ | $10.0_{0.1}$ | $7.8_{0.1}$ | $9.0_{0.0}$ | $10.2_{0.0}$ | $10.0_{0.0}$ | $42.0_{0.1}$ | $44.0_{0.1}$ | $40.4_{0.1}$ | $47.2_{0.1}$ | $43.1_{0.2}$ | $39.3_{0.1}$ | $44.9_{0.2}$ |
| Base+VE | $10.7_{0.0}$ | $10.4_{0.1}$ | $10.5_{0.0}$ | $10.4_{0.0}$ | $10.5_{0.0}$ | $10.7_{0.0}$ | $10.6_{0.0}$ | $33.8_{0.1}$ | $37.7_{0.1}$ | $36.4_{0.1}$ | $40.1_{0.1}$ | $36.3_{0.2}$ | $36.2_{0.2}$ | $38.9_{0.0}$ |
| CV | $11.6_{0.1}$ | $11.2_{0.1}$ | $11.4_{0.0}$ | $10.9_{0.0}$ | $11.0_{0.0}$ | $11.2_{0.0}$ | $10.8_{0.1}$ | $32.3_{0.1}$ | $33.8_{0.3}$ | $30.2_{0.1}$ | $34.1_{0.1}$ | $30.4_{0.1}$ | $32.5_{0.1}$ | $31.6_{0.1}$ |
| Chat | $14.0_{0.0}$ | $14.0_{0.0}$ | $14.0_{0.0}$ | $14.0_{0.0}$ | $14.0_{0.0}$ | $14.0_{0.0}$ | $14.0_{0.0}$ | $27.4_{0.1}$ | $54.9_{0.1}$ | $34.1_{0.1}$ | $55.2_{0.2}$ | $45.7_{0.1}$ | $50.8_{0.0}$ | $54.7_{0.2}$ |
| Chat+CPT | $10.6_{0.0}$ | $10.9_{0.1}$ | $10.4_{0.1}$ | $10.6_{0.0}$ | $12.0_{0.0}$ | $11.2_{0.0}$ | $11.1_{0.0}$ | $29.8_{0.1}$ | $37.4_{0.2}$ | $35.1_{0.4}$ | $38.4_{0.3}$ | $39.4_{0.1}$ | $37.2_{0.2}$ | $41.3_{0.3}$ |
| Chat+VE | $11.5_{0.0}$ | $11.4_{0.0}$ | $11.6_{0.0}$ | $10.9_{0.0}$ | $11.5_{0.0}$ | $11.3_{0.1}$ | $11.7_{0.1}$ | $30.1_{0.3}$ | $34.7_{0.1}$ | $31.9_{0.1}$ | $35.7_{0.2}$ | $31.6_{0.1}$ | $34.8_{0.0}$ | $33.0_{0.1}$ |
| ElChat | $12.1_{0.0}$ | $12.0_{0.1}$ | $11.9_{0.0}$ | $12.1_{0.1}$ | $12.6_{0.1}$ | $12.5_{0.1}$ | $12.3_{0.0}$ | $26.0_{0.1}$ | $26.3_{0.3}$ | $31.0_{0.2}$ | $33.8_{0.0}$ | $23.8_{0.4}$ | $30.8_{0.4}$ | $32.6_{0.2}$ |
| ElChat \Merge | $11.3_{0.0}$ | $11.2_{0.0}$ | $11.4_{0.0}$ | $10.4_{0.0}$ | $11.1_{0.0}$ | $9.3_{0.1}$ | $11.1_{0.1}$ | $26.7_{0.2}$ | $36.8_{0.1}$ | $33.5_{0.1}$ | $36.2_{0.2}$ | $34.4_{0.1}$ | $35.4_{0.2}$ | $34.0_{0.1}$ |
| ElChat \Copy | $12.2_{0.1}$ | $12.0_{0.1}$ | $11.8_{0.0}$ | $12.1_{0.0}$ | $12.1_{0.1}$ | $12.0_{0.0}$ | $12.1_{0.1}$ | $26.2_{0.3}$ | $25.9_{0.2}$ | $31.4_{0.3}$ | $34.6_{0.2}$ | $26.5_{0.2}$ | $29.6_{0.1}$ | $30.3_{0.2}$ |
| ElChat (L) | $12.1_{0.1}$ | $12.0_{0.1}$ | $11.8_{0.1}$ | $12.1_{0.0}$ | $12.5_{0.0}$ | $12.5_{0.0}$ | $12.3_{0.0}$ | $26.3_{0.2}$ | $25.3_{0.1}$ | $30.8_{0.3}$ | $34.3_{0.3}$ | $22.8_{0.3}$ | $30.7_{0.1}$ | $32.2_{0.2}$ |

Table 16: Qwen3 14B task performance. Darker blue and red shades indicate higher positive and negative relative performance change over  Chat  per language and task, respectively.

| | Target sum [ROUGE-L] | | | English → Target mt [chrF++] | | |
|---|---|---|---|---|---|---|
| | am | bn | te | am | bn | te |
| Base | $2.5_{0.0}$ | $0.0_{0.0}$ | $2.4_{0.0}$ | $11.7_{0.3}$ | $38.3_{0.1}$ | $32.6_{0.1}$ |
| Base+CPT | $2.2_{0.0}$ | $0.0_{0.0}$ | $2.4_{0.1}$ | $25.8_{0.1}$ | $39.3_{0.0}$ | $34.6_{0.2}$ |
| Base+VE | $2.6_{0.2}$ | $0.0_{0.0}$ | $3.4_{0.1}$ | $17.9_{0.1}$ | $29.3_{0.2}$ | $29.6_{0.2}$ |
| CV | $4.1_{0.2}$ | $0.0_{0.0}$ | $3.8_{0.1}$ | $21.7_{0.1}$ | $30.3_{0.1}$ | $30.3_{0.1}$ |
| Chat | $2.3_{0.0}$ | $0.0_{0.0}$ | $3.0_{0.0}$ | $11.8_{0.1}$ | $37.2_{0.0}$ | $30.1_{0.0}$ |
| Chat+CPT | $2.9_{0.1}$ | $0.0_{0.0}$ | $2.6_{0.0}$ | $23.6_{0.1}$ | $37.7_{0.1}$ | $32.9_{0.1}$ |
| Chat+VE | $3.7_{0.2}$ | $0.0_{0.0}$ | $3.1_{0.0}$ | $22.8_{0.2}$ | $30.2_{0.1}$ | $28.2_{0.2}$ |
| ElChat | $3.8_{0.1}$ | $0.0_{0.0}$ | $3.0_{0.0}$ | $25.3_{0.3}$ | $31.1_{0.2}$ | $28.0_{0.0}$ |
| | **English sum [ROUGE-L]** | | | **Target → English mt [chrF++]** | | |
| | am | bn | te | am | bn | te |
| Base | $12.0_{0.0}$ | $12.0_{0.0}$ | $12.0_{0.0}$ | $37.6_{0.1}$ | $58.5_{0.0}$ | $58.8_{0.2}$ |
| Base+CPT | $11.4_{0.0}$ | $11.3_{0.1}$ | $11.1_{0.0}$ | $45.5_{0.2}$ | $47.2_{0.0}$ | $48.9_{0.3}$ |
| Base+VE | $11.8_{0.0}$ | $11.0_{0.0}$ | $11.6_{0.0}$ | $37.4_{0.2}$ | $36.7_{0.0}$ | $34.0_{0.2}$ |
| CV | $10.8_{0.0}$ | $10.4_{0.0}$ | $10.7_{0.0}$ | $43.4_{0.2}$ | $30.5_{0.1}$ | $35.5_{0.1}$ |
| Chat | $11.5_{0.0}$ | $11.5_{0.0}$ | $11.5_{0.0}$ | $36.6_{0.0}$ | $57.3_{0.0}$ | $57.4_{0.0}$ |
| Chat+CPT | $11.4_{0.0}$ | $11.5_{0.0}$ | $11.4_{0.0}$ | $53.0_{0.0}$ | $56.4_{0.0}$ | $56.6_{0.0}$ |
| Chat+VE | $11.5_{0.0}$ | $11.2_{0.0}$ | $11.5_{0.0}$ | $49.8_{0.1}$ | $48.5_{0.0}$ | $39.9_{0.3}$ |
| ElChat | $11.5_{0.0}$ | $11.4_{0.0}$ | $11.7_{0.0}$ | $51.1_{0.1}$ | $49.0_{0.1}$ | $45.4_{0.1}$ |

Table 17: Qwen2.5 7B inference speedup measured by the number of tokens generated per second. Darker blue and red shades indicate higher positive and negative relative performance change over  Chat  per language and task, respectively.

| | Target sum | | | | | | | English → Target mt | | | | | | | Target mc | | | | | | |
|---|---|---|---|---|---|---|---|---|---|---|---|---|---|---|---|---|---|---|---|---|---|
| | am | bn | my | gu | si | ta | te | am | bn | my | gu | si | ta | te | am | bn | my | gu | si | ta | te |
| Base | $29.3_{0.3}$ | $30.7_{0.2}$ | $27.8_{0.2}$ | $27.2_{0.1}$ | $32.4_{0.1}$ | $31.6_{0.2}$ | $27.6_{0.1}$ | $34.1_{0.3}$ | $33.4_{0.4}$ | $33.2_{0.2}$ | $33.9_{0.2}$ | $36.4_{0.3}$ | $33.9_{0.1}$ | $34.0_{0.3}$ | 28.9 | 22.6 | 15.7 | 17.5 | 18.2 | 20.9 | 17.1 |
| Base+CPT | $30.7_{1.1}$ | $26.6_{2.0}$ | $28.8_{0.2}$ | $28.1_{0.1}$ | $28.0_{1.6}$ | $33.1_{0.3}$ | $29.7_{0.2}$ | $35.5_{0.1}$ | $35.4_{0.2}$ | $34.6_{0.0}$ | $35.0_{0.1}$ | $36.3_{0.3}$ | $35.3_{0.3}$ | $35.5_{0.1}$ | 28.5 | 23.1 | 15.2 | 17.1 | 18.6 | 20.7 | 18.0 |
| Base+VE | $110.4_{1.0}$ | $68.9_{1.2}$ | $103.1_{0.1}$ | $106.3_{0.3}$ | $102.3_{0.3}$ | $70.8_{0.7}$ | $89.9_{0.8}$ | $86.5_{0.3}$ | $72.2_{0.7}$ | $93.5_{0.7}$ | $98.0_{1.4}$ | $102.7_{0.8}$ | $73.6_{0.4}$ | $91.3_{1.0}$ | 38.1 | 37.4 | 40.2 | 42.6 | 37.7 | 36.1 | 37.3 |
| CV | $121.8_{0.3}$ | $76.2_{0.2}$ | $98.9_{0.6}$ | $93.4_{0.4}$ | $114.7_{1.6}$ | $80.8_{0.7}$ | $89.8_{0.6}$ | $109.6_{0.5}$ | $71.6_{0.5}$ | $97.5_{1.1}$ | $94.5_{0.6}$ | $70.4_{0.4}$ | $67.2_{0.4}$ | $82.3_{0.3}$ | 40.8 | 42.2 | 37.1 | 37.3 | 41.5 | 39.4 | 35.4 |
| Chat | $30.4_{0.3}$ | $30.7_{0.0}$ | $29.6_{0.1}$ | $27.3_{0.1}$ | $29.8_{0.3}$ | $33.3_{0.2}$ | $29.6_{0.1}$ | $32.6_{0.1}$ | $34.7_{0.3}$ | $34.7_{0.3}$ | $35.3_{0.2}$ | $37.9_{0.3}$ | $35.2_{0.1}$ | $35.9_{0.3}$ | 27.9 | 22.0 | 15.3 | 16.9 | 18.0 | 20.2 | 17.0 |
| Chat+CPT | $33.7_{0.2}$ | $32.2_{0.2}$ | $29.4_{0.1}$ | $27.3_{0.0}$ | $33.5_{0.0}$ | $33.3_{0.1}$ | $28.2_{0.0}$ | $34.8_{0.4}$ | $34.6_{0.3}$ | $33.7_{0.2}$ | $34.2_{0.1}$ | $35.3_{0.1}$ | $34.3_{0.1}$ | $34.9_{0.1}$ | 28.8 | 23.8 | 15.5 | 17.2 | 19.7 | 21.0 | 17.5 |
| Chat+VE | $129.1_{0.7}$ | $68.3_{0.0}$ | $103.6_{0.6}$ | $94.8_{0.3}$ | $104.3_{0.7}$ | $80.6_{0.7}$ | $99.4_{0.4}$ | $102.0_{1.8}$ | $73.3_{0.6}$ | $97.6_{0.0}$ | $98.6_{0.8}$ | $81.1_{0.9}$ | $70.9_{0.7}$ | $80.2_{0.9}$ | 43.2 | 37.5 | 39.1 | 37.6 | 37.3 | 40.2 | 38.7 |
| ElChat | $117.5_{0.4}$ | $76.8_{0.8}$ | $97.6_{5.0}$ | $95.2_{0.4}$ | $104.2_{0.2}$ | $81.1_{0.8}$ | $99.7_{1.0}$ | $113.3_{0.5}$ | $70.9_{0.5}$ | $94.1_{0.4}$ | $93.0_{0.9}$ | $101.1_{0.2}$ | $74.2_{0.8}$ | $91.6_{0.9}$ | 38.0 | 42.0 | 37.2 | 37.4 | 37.7 | 39.9 | 38.4 |

Table 18: Llama 3.1 8B inference speedup measured by the number of tokens generated per second. Darker blue and red shades indicate higher positive and negative relative performance change over  Chat  per language and task, respectively. (L) stands for linear merging.

| | Target sum | | | | | | | English → Target mt | | | | | | | Target mc | | | | | | |
|---|---|---|---|---|---|---|---|---|---|---|---|---|---|---|---|---|---|---|---|---|---|
| | am | bn | my | gu | si | ta | te | am | bn | my | gu | si | ta | te | am | bn | my | gu | si | ta | te |
| Base | $19.3_{0.0}$ | $21.7_{0.0}$ | $18.3_{0.1}$ | $19.7_{0.0}$ | $20.8_{0.0}$ | $22.4_{0.1}$ | $18.8_{0.0}$ | $31.9_{0.0}$ | $33.3_{0.1}$ | $28.5_{0.1}$ | $32.0_{0.0}$ | $31.9_{0.0}$ | $32.1_{0.0}$ | $31.5_{0.2}$ | 15.4 | 17.7 | 10.5 | 13.5 | 13.1 | 14.6 | 12.7 |
| Base+CPT | $19.3_{0.0}$ | $21.9_{0.0}$ | $18.5_{0.0}$ | $19.4_{0.1}$ | $21.2_{0.0}$ | $23.2_{0.0}$ | $18.9_{0.0}$ | $31.6_{0.0}$ | $33.1_{0.4}$ | $28.4_{0.2}$ | $31.6_{0.1}$ | $31.6_{0.1}$ | $31.8_{0.1}$ | $31.4_{0.2}$ | 15.3 | 17.9 | 10.5 | 13.6 | 13.1 | 14.7 | 12.7 |
| Base+VE | $206.2_{3.3}$ | $78.5_{0.5}$ | $114.7_{0.3}$ | $100.6_{0.5}$ | $121.0_{0.4}$ | $91.4_{0.4}$ | $105.4_{0.2}$ | $141.5_{0.8}$ | $78.9_{0.5}$ | $111.6_{0.4}$ | $98.2_{0.8}$ | $122.5_{1.2}$ | $84.9_{0.8}$ | $98.3_{0.8}$ | 44.7 | 39.7 | 36.9 | 36.9 | 37.0 | 37.8 | 38.7 |
| CV | $224.3_{2.9}$ | $78.2_{0.1}$ | $120.2_{0.2}$ | $107.6_{0.6}$ | $132.7_{0.1}$ | $90.1_{1.1}$ | $103.5_{0.2}$ | $185.1_{0.8}$ | $84.8_{0.4}$ | $123.5_{1.0}$ | $135.6_{1.4}$ | $123.2_{1.0}$ | $91.8_{1.1}$ | $109.5_{1.2}$ | 44.5 | 33.0 | 28.6 | 33.4 | 34.7 | 31.0 | 31.4 |
| Chat | $19.2_{0.0}$ | $22.0_{0.0}$ | $18.3_{0.0}$ | $19.6_{0.0}$ | $21.1_{0.1}$ | $22.9_{0.0}$ | $18.9_{0.0}$ | $31.6_{0.1}$ | $33.0_{0.2}$ | $28.2_{0.0}$ | $31.5_{0.1}$ | $31.7_{0.0}$ | $31.6_{0.1}$ | $31.0_{0.1}$ | 15.0 | 17.3 | 10.3 | 13.2 | 12.9 | 14.4 | 12.5 |
| Chat+CPT | $18.9_{0.0}$ | $21.8_{0.0}$ | $18.4_{0.0}$ | $19.5_{0.0}$ | $21.1_{0.0}$ | $23.2_{0.1}$ | $18.9_{0.0}$ | $31.8_{0.1}$ | $33.4_{0.1}$ | $28.3_{0.0}$ | $31.7_{0.2}$ | $31.3_{0.3}$ | $31.9_{0.1}$ | $31.3_{0.1}$ | 14.9 | 17.2 | 10.4 | 13.3 | 13.0 | 14.4 | 12.7 |
| Chat+VE | $218.9_{3.1}$ | $78.6_{0.3}$ | $120.2_{0.7}$ | $107.2_{0.5}$ | $135.1_{0.4}$ | $93.2_{0.6}$ | $99.3_{0.5}$ | $168.8_{3.7}$ | $79.7_{0.2}$ | $115.5_{1.2}$ | $100.4_{1.5}$ | $124.4_{0.9}$ | $89.0_{0.5}$ | $96.9_{0.4}$ | 44.3 | 39.1 | 34.7 | 39.5 | 40.6 | 37.6 | 36.9 |
| ElChat | $233.1_{9.9}$ | $75.8_{0.5}$ | $118.2_{0.6}$ | $105.5_{0.5}$ | $126.3_{0.6}$ | $90.3_{1.4}$ | $101.7_{0.4}$ | $212.0_{3.4}$ | $81.3_{0.8}$ | $113.0_{1.2}$ | $107.7_{0.4}$ | $124.1_{0.7}$ | $89.8_{0.8}$ | $107.2_{0.5}$ | 44.5 | 37.9 | 34.1 | 38.9 | 38.0 | 37.6 | 37.3 |
| ElChat \Merge | $201.3_{2.6}$ | $79.3_{0.0}$ | $121.6_{0.4}$ | $109.1_{0.2}$ | $132.1_{1.1}$ | $88.9_{0.4}$ | $103.8_{0.2}$ | $164.1_{2.2}$ | $76.3_{0.8}$ | $111.4_{1.5}$ | $84.2_{1.1}$ | $102.0_{2.7}$ | $84.3_{0.4}$ | $94.5_{0.3}$ | 42.2 | 39.1 | 34.4 | 40.2 | 39.9 | 37.3 | 37.6 |
| ElChat \Copy | $235.0_{4.3}$ | $79.2_{0.1}$ | $118.6_{0.2}$ | $107.3_{0.2}$ | $123.7_{0.7}$ | $85.6_{1.5}$ | $105.0_{0.6}$ | $207.3_{1.6}$ | $83.7_{1.4}$ | $113.2_{0.7}$ | $112.1_{0.3}$ | $129.2_{0.5}$ | $92.4_{0.8}$ | $110.9_{2.0}$ | 46.7 | 39.6 | 34.3 | 39.5 | 37.5 | 36.4 | 37.8 |
| ElChat (L) | $230.7_{2.4}$ | $78.7_{0.2}$ | $119.6_{0.6}$ | $106.3_{0.4}$ | $135.9_{0.2}$ | $92.6_{0.9}$ | $103.9_{0.8}$ | $211.4_{3.1}$ | $78.9_{0.7}$ | $111.2_{1.1}$ | $106.0_{0.7}$ | $122.2_{0.5}$ | $88.4_{1.9}$ | $105.8_{0.5}$ | 44.9 | 39.2 | 34.3 | 39.4 | 40.8 | 37.7 | 37.8 |

Table 19: Qwen3 14B inference speedup measured by the number of tokens generated per second. Darker blue and red shades indicate higher positive and negative relative performance change over  Chat  per language and task, respectively.

| | Target sum | | | English $\rightarrow$ Target mt | | | Target mc | | |
|---|---|---|---|---|---|---|---|---|---|
| | am | bn | te | am | bn | te | am | bn | te |
| Base | $14.6_{0.1}$ | $13.5_{0.1}$ | $13.2_{0.1}$ | $15.4_{0.1}$ | $15.6_{0.2}$ | $15.6_{0.1}$ | 17.7 | 16.3 | 12.6 |
| Base+CPT | $14.6_{0.0}$ | $14.6_{0.1}$ | $13.8_{0.1}$ | $15.5_{0.2}$ | $16.1_{0.2}$ | $15.4_{0.2}$ | 17.5 | 16.6 | 12.5 |
| Base+VE | $52.3_{0.2}$ | $33.7_{0.5}$ | $43.5_{0.2}$ | $36.3_{0.3}$ | $29.6_{0.5}$ | $38.6_{0.3}$ | 17.8 | 17.5 | 17.8 |
| CV | $56.8_{0.2}$ | $34.3_{0.1}$ | $44.0_{0.4}$ | $27.9_{0.1}$ | $26.3_{0.2}$ | $35.2_{0.4}$ | 17.0 | 17.6 | 17.5 |
| Chat | $15.2_{0.2}$ | $14.6_{0.1}$ | $14.1_{0.1}$ | $15.7_{0.1}$ | $15.9_{0.2}$ | $15.8_{0.2}$ | 17.9 | 16.3 | 12.3 |
| Chat+CPT | $14.9_{0.1}$ | $14.5_{0.1}$ | $14.1_{0.0}$ | $16.0_{0.0}$ | $15.9_{0.2}$ | $15.8_{0.1}$ | 17.4 | 15.9 | 12.3 |
| Chat+VE | $55.8_{0.2}$ | $34.3_{0.3}$ | $44.9_{0.1}$ | $28.1_{0.3}$ | $32.3_{0.3}$ | $42.8_{0.7}$ | 17.4 | 17.6 | 17.6 |
| ElChat | $58.0_{0.5}$ | $34.5_{0.1}$ | $44.1_{0.2}$ | $43.5_{0.7}$ | $30.9_{0.5}$ | $40.3_{0.5}$ | 17.4 | 17.6 | 17.8 |

