# OpenReview forum: "Adapting Chat Language Models Using Only Target Unlabeled Language Data"
_TMLR — Accepted by TMLR_

### Review · Reviewer_nw3R · 2025-06-03

**Summary Of Contributions:**

To adding new tokens and continuing pre-training on LLMs for target data, in this paper, the authors try to first adapt the base LLM with VE (vocabulary expansion) on target unlabeled data. The proposed method then converts the base LLM with VE to a chat LLM by adding a Chat Vector (CV) according to the weight difference between the source base LLM and chat LLMs. The proposed method, ElChat, performs more robust and competitive on target language and excel strong baselines.

**Audience:**

Yes

**Claims And Evidence:**

Yes

**Requested Changes:**

See the weakness

**Strengths And Weaknesses:**

Pros:

1.It is interesting to explore a new perspective of adapting the base LLM with VE (vocabulary expansion) on target unlabeled data.

2.Vocabulary expansion is an effective method for adapting to training on unlabeled data. Vocabulary expansion is a good point for doing well on target unlabeled data.

3.The proposed method makes sense in this setting and also perform well on several benchmarks.


Cons:

1.About the description, “chat model” seems not so distinct with the so called LLM. What the core difference between the two kinds of models? Which point in the differences between chat model and LLMs make that we need to customize a new kinds of method. This paper mainly talks about chat model, so this paper should make them clear and also make it clear that the key contributions of the proposed method are indeed related to the chat model.

2.As LLM is known trained on a very large scaled data, the vocabulary may be already large enough. In this way, do is still necessary to expand the vocabulary?

3.As for the core operation, “VE”, it is not so clear about how to process VE. The authors mentioned that expanding its input and output head matrices with new tokens. But which part is the input head matrices? Which part is the output head matrices? How to expand with new tokens? The s specific roles (e.g. user, system, or assistant) is also not so clear. Why do we need a user/system/assistant in VE. VE is just to expand the vocabulary. How to expand the vocabulary with the user/system/assistant? The authors may provide some details in the appendix, but in the paper body, it is necessary to make the method clear to understand.

4.The technical novelty reflected in the section 3 is also limited. VE is something about expanding matrices and prompt engineering (also cannot get any insights from the prompt engineering). Source and Target Chat Model Merging comes from existing papers.

---

> ### Author Response · Authors · 2025-06-12
> **Thanks for your constructive review**
>
> We appreciate your constructive feedback. Below, we provide our point-to-point responses.
>
> **On the contributions of the paper**: We appreciate that you find our method robust and sound. It is crucial to clarify that, unlike Chat Vector, our proposed method, ElChat, does not require access to the base model. As visualized in Figure 1, this is a critical distinction because (i) base models are not always available due to reasons such as safety; and (ii) it is completely at the discretion of model developers whether they publish both base and chat variants (refer to page 1; Introduction; paragraph 3). This independence contributes to ElChat's broader applicability.
>
> ---
>
> **On the difference between base and chat models and ElChat's relevance**: We appreciate you raising this point. To clarify, in our paper, a **base model** refers to an LLM pre-trained on unlabeled data without any further post-training. A **chat model**, on the other hand, is a base model that has been further supervised fine-tuned on labeled conversational data, enabling it to follow instructions. Note that we already mentioned this distinction in the Introduction, paragraph 2 on page 1. We will make this more clear in the revised version of the paper.
>
> The motivation for both Chat Vector and ElChat is that labeled conversational data required for target-language chat model adaptation is not always available or is very costly for low-resource languages. This challenge directly motivates our contributions. We proposed ElChat because the prior method, Chat Vector, relies on access to a base model from the same family, which is not always available as highlighted above. ElChat overcomes this by using model merging and copying special token weights to bypass the need for the base model. Thus, the core contributions of our proposed method are indeed specifically related to addressing the unique adaptation challenges of chat models.
>
> ---
>
> **On the necessity of vocabulary expansion**: Thanks for raising this great question.  While it is true that the vocabularies of LLMs often are large, e.g., 152K for Qwen2.5 and 128K for Llama 3.1, they still suffer from tokenization overfragmentation in underrepresented languages. This means such languages often require substantially more inference steps than their high-resource counterparts. For instance, as described in the Introduction, paragraph 1 on page 1, one of our target languages, Amharic, requires 3.48x more inference steps in Qwen2.5 without vocabulary expansion.
>
> This is precisely why vocabulary expansion is so crucial. As shown in Figure 4 on page 8, vocabulary expansion contributes to at least 2x speedups for our target languages in both Qwen2.5 and Llama 3.1. These underscore the continued importance of vocabulary expansion even for models with large vocabularies.
>
> ---
>
> **On the vocabulary expansion process**: The input matrix is the token embedding matrix, and the output head matrix is the language modeling head (linear projection layer). We expand these by following the standard protocol adopted in Tejaswi et al. (2024) and Mundra et al. (2024) by resizing and initializing new token embeddings/weights.
>
> Regarding roles (e.g., user, system, assistant) in the second paragraph on page 4, these relate to chat templates and continual pre-training (CPT), not VE directly. We acknowledge this placement might have caused confusion, and this explanation will be moved under a distinct paragraph to enhance clarity in the revised paper.
>
> Atula Tejaswi, Nilesh Gupta, and Eunsol Choi. 2024. Exploring Design Choices for Building Language-Specific LLMs. In Findings of the Association for Computational Linguistics: EMNLP 2024
> Nandini Mundra, Aditya Nanda Kishore Khandavally, Raj Dabre, Ratish Puduppully, Anoop Kunchukuttan, and Mitesh M Khapra. 2024. An Empirical Comparison of Vocabulary Expansion and Initialization Approaches For Language Models. In Proceedings of the 28th Conference on Computational Natural Language Learning
>
> ---
>
> **On the technical novelty of ElChat**: We acknowledge that each component of ElChat can be seen as an established technique. However, their combination and application within the context of target-language chat model adaptation represents an important technical contribution and novelty. This is indeed clearly demonstrated in our ablation analysis in Table 2. This table shows that removing either of the two key components (model merging and weight copying) substantially reduces performance across tasks. This highlights their complementarity and non-trivial synergistic effect in eliciting ElChat’s abilities, and their combination is essential for obtaining strong performance. We will further stress this in the revised version of the paper.
>
> ---
>
> Thank you again for the constructive feedback. Finally, we will update the paper to reflect the above responses so that we can make the paper more concise and easy-to-understand, once the other two reviews have been posted.

---

> > ### Comment · Reviewer_nw3R · 2025-08-10
> > **Further comments**
> >
> > The authors mainly addressed my concerns.
> >
> > Please make sure to:
> >
> > 1. explain more about the definition of the so call "chat model" in the paper.
> >
> > 2. explain more about the motivation about VE, given the fact that most LLMs have a large vocabulary.
> >
> > About the above two issues, I see the authors' explanations in the rebuttal but it would be better to explain them in the paper.

---

> > > ### Author Response · Authors · 2025-08-10
> > >
> > > Thanks for your response.
> > >
> > > On the first point, we have already incorporated your suggestion into the revised manuscript, as mentioned in the revision note. Specifically, the changes can be found in Figure 1's caption and the second paragraph on page 1.
> > >
> > > On the second point, we agree that a clearer explanation can be helpful. While we previously discussed this in the first paragraph on page 1, we will be uploading a new revision to specifically highlight that even models with large vocabularies still suffer from overfragmentation in underrepresented languages, making VE a necessary approach.

---

### Review · Reviewer_f5vZ · 2025-06-15

**Summary Of Contributions:**

This paper introduces a novel method for adapting Large Language Models (LLMs). The proposed approach involves conducting Continued Pre-training (CPT) directly on a chat model, followed by merging the weights of the original chat model with the newly trained one. A key contribution of this work is that it eliminates the reliance on base models, a common requirement for previous adaption techniques. The experimental results indicate that the adapted model achieves significant improvements in both performance on downstream tasks in less-resourced languages and in inference speed.

**Audience:**

Yes

**Claims And Evidence:**

Yes

**Requested Changes:**

In addition to the weaknesses mentioned above, I believe the authors should provide an explanation for a somewhat counter-intuitive result observed in Figure 2(a). Specifically, it shows that the fine-tuned model achieves significantly better performance than the original chat model on the English-language versions of the TruthfulQA, ToxiGen, and ImplicitHate datasets. This outcome is unexpected and warrants further clarification."

**Strengths And Weaknesses:**

Overall, the motivation of this paper is very clear, and the work holds significant potential value for improving the fairness of LLM applications. The methodology employed is generally direct, and its effectiveness appears to be quite significant, as it demonstrably improves the performance of various LLMs on tasks in low-resource languages.

However, the paper has several weaknesses, primarily in the following three areas:

(1) Insufficient Experimentation: The experiments are limited, as the paper only compares models at the ~8B parameter scale. While I understand that conducting Continued Pre-training (CPT) on larger LLMs may pose practical challenges for academic institutions, this remains a weakness. Furthermore, the study does not report the performance of any recent state-of-the-art (SOTA) models (such as Llama 4 or Qwen 3) on the same tasks. This omission makes it difficult to contextualize the current progress in solving this problem.

(2) Overly Brief Methodological Description: The description of the method is too simplistic, preventing readers from fully understanding the implementation details. The methodology section occupies less than a page, and many crucial details that should have been explained in the paper are instead briefly mentioned via references. Compounded by the lack of open-sourced code, the paper feels more like a high-level overview of a technical approach rather than a detailed study, which hinders reader comprehension.

(3) Limited Novelty of the Method: Frankly, the method itself seems overly simple and appears to be merely a straightforward combination of existing techniques rather than a novel contribution.

---

> ### Author Response · Authors · 2025-06-18
> **Thanks for your thorough review**
>
> Thanks for your thorough review. Below, let us provide point-to-point responses:
>
> **On the insufficient experimentation**: We agree that it is worthwhile testing the proposed method on a larger model. However, as you kindly noted, computational constraints are a significant practical challenge for us. To this end, we prioritized the experiments on similar model sizes from different model families (i.e. Qwen2.5 7B and Llama 3.1 8B) to examine the robustness of the approach rather than trying different model scales. While these scales are not the largest models available, they have been widely used in recent work on language adaptation  (e.g. [1], [2], [3]).
>
> [1] Trans-Tokenization and Cross-lingual Vocabulary Transfers: Language Adaptation of LLMs for Low-Resource NLP (Remy et al., COLM 2024)
> [2] Exploring Design Choices for Building Language-Specific LLMs (Tejaswi et al., EMNLP Findings 2024)
> [3] An Empirical Comparison of Vocabulary Expansion and Initialization Approaches For Language Models (Mundra et al., CoNLL 2024)
>
> Regarding the omission of models like Llama 4 or Qwen 3, it is important to clarify the timeline: these models were released in April 2025, shortly before our TMLR submission date of May 16th, 2025. At that point, the vast majority of our experiments were already complete, posing significant technical and logistical challenges to re-run everything with new backbones. Nonetheless, we acknowledge the importance of using SoTA models. If resources become available, we are keen to provide additional results with some of these models.
>
> ---
>
> **On the methodological description**: We understand the concern about the briefness of the methodology section. To remedy this, we will make the following revisions once all reviews become available:
> * We will expand each named paragraph (i.e., VE, Merge, and Copy) by including granular, step-by-step explanations with mathematical notations.
> * We will include an overview figure describing each process visually.
>
> It is worth noting that we have uploaded anonymized code as supplementary material on the OpenReview paper forum page, and we will, of course, publish the code and models upon publication to facilitate reproducibility.
>
> ---
>
> **On the novelty of the proposed approach**: We acknowledge that each component of ElChat can be seen as an established technique. However, their combination and application within the context of target-language chat model adaptation represents an important technical contribution and novelty. This is indeed clearly demonstrated in our ablation analysis in Table 2. This table shows that removing either of the two key components (model merging and weight copying) substantially reduces performance across tasks. This highlights their complementarity and non-trivial synergistic effect in eliciting ElChat’s abilities, and their combination is essential for obtaining strong performance. We will further stress this in the revised version of the paper.
>
> ---
>
> **On the performance of TruthfulQA, ToxiGen, and ImplicitHate**: Thanks for asking this question. Our evaluation on the three datasets is indeed not in English but in target languages. This setup follows Cahyawijaya et al. (2024) and uses the translated data. We will make this more explicit in the revised version of the paper.
>
> Cendol: Open instruction-tuned generative large language models for Indonesian languages (Cahyawijaya et al., ACL 2024)

---

### Review · Reviewer_BFxi · 2025-06-24

**Summary Of Contributions:**

This paper introduces ElChat, a method for adapting chat-based large language models (LLMs) to low-resource target languages without requiring access to a base model or labeled chat data. ElChat builds on vocabulary expansion (VE) by applying it directly to the source chat model and then recovering lost instruction-following capabilities through model merging (via SLERP) and special token weight copying. The approach is evaluated across two chat models (Qwen2.5 and Llama 3.1), seven typologically diverse languages, and a wide range of tasks including instruction following, safety, and multilingual understanding. The paper compares ElChat against strong baselines including Chat Vector (CV) and Chat+VE, and includes ablations and inference efficiency analyses.

**Audience:**

Yes

**Broader Impact Concerns:**

While the paper does not raise immediate or serious broader impact concerns, it would benefit from a more explicit discussion on bias transfer, safety degradation, and potential misuse in low-resource language settings. These risks are especially relevant given the growing accessibility of adapted chat LLMs in under-monitored regions.

**Claims And Evidence:**

Yes

**Requested Changes:**

- Explicitly state in the paper that the novelty lies in the combination of known techniques under practical constraints, rather than in individual components.
- Include more robust instruction-following or multi-turn dialogue evaluation in target languages.
- Include results on AlpacaEval, Chatbot Arena, or similar frameworks to validate instruction-following behavior more holistically.
- Add discussion or per-task breakdowns on where ElChat meaningfully outperforms CV, and where it does not.
- Provide a rationale or deeper empirical comparison between SLERP and linear interpolation, especially since the latter sometimes performs as well or better.

**Strengths And Weaknesses:**

Strengths:
- Addresses a realistic low-resource setting where base models or translated chat data are not available.
- Leverages established techniques (VE, SLERP, token copying) in a well-integrated pipeline.
- Includes instruction-following, safety, source/target language tasks, and inference speed.
- Covers seven non-Latin-script, underrepresented languages using curated adaptation data.
- Shows that merging and token copying each contribute meaningfully to final performance.

Weaknesses:
- ElChat is a combination of existing techniques rather than a fundamentally new algorithm or method. SLERP and token-copying are reused without innovation; the novelty lies in their integration only.
- ElChat’s improvements over the Chat Vector baseline are modest and task-dependent. In some safety tasks (e.g., ToxiGen), and discriminative classification tasks (e.g., gmmlu), CV actually outperforms ElChat.
- Only MGSM is used for multilingual reasoning, which is narrow in scope and covers just two languages.
- No use of generalization/stress-test benchmarks like BigBench Hard (BBH).
- Linear merging (ElChat-L) performs comparably in some tasks, yet SLERP is used by default without deeper analysis.

---

> ### Author Response · Authors · 2025-06-26
> **Thank you for your valuable feedback.**
>
> Thank you for your valuable feedback. We provide our point-to-point responses in the following.
>
> **On the novelty of ElChat**: We agree with the reviewer's assessment that the novelty of ElChat lies in its unique combination and application of established techniques (model merging and copying) to address practical constraints, rather than in the individual components themselves. To address the request for clarity, we will add an explicit statement in Section 3, in the introductory paragraphs, clearly stating that the novelty of ElChat resides in its strategic combination of known techniques to overcome specific challenges, rather than in the individual components. This will build upon our existing mention on page 9 in the Ablation paragraph.
>
> ---
>
> **On the target-language instruction-following and multi-turn dialogue evaluation**: We completely agree on the importance of conducting instruction-following and multi-turn dialogue evaluation in target languages. Nonetheless, as we mentioned in the third paragraph on page 5 (4.5 Evaluation Tasks) and the Limitations section, a significant challenge lies in the current scarcity of established instruction-following and multi-turn dialogue benchmarks that specifically include our low-resource target languages. For instance, while multiple multilingual instruction-following benchmarks have been proposed recently (e.g. Multi-IF, XIFBench, and M-IFEval), none of these appear to include our low-resource target languages. We hope this study will motivate the development of instruction-following and multi-turn dialogue benchmarks in low-resource languages in the future.
>
> Multi-IF: https://arxiv.org/abs/2410.15553 covers French, Russian, Hindi, Italian, Portuguese, Spanish and Chinese.
> XIFBench: https://arxiv.org/abs/2503.07539 covers English, Chinese, Russian, Arabic, Hindi, and Swahili.
> M-IFEval: https://aclanthology.org/2025.findings-naacl.344/ covers French, Japanese, and Spanish.
>
> ---
>
> **On the additional evaluation on generalization and English instruction-following benchmarks**: Thanks for pointing this out. We acknowledge the importance of conducting more holistic evaluation on these aspects. To this end, we are now preparing to conduct evaluations on BigBench Hard and AlpacaEval as additional analysis. Note that due to resource constraints, the corresponding evaluation will use a subset of available models and languages. We will add the results to Section 5 in the revised version of the paper.
>
> ---
>
> **On the discussion comparing ElChat and Chat Vector**: We understand the need for a clearer discussion on where ElChat meaningfully outperforms Chat Vector (CV), and where it does not.
>
> As mentioned in the second item of the contributions on page 2, our experiments demonstrate two key trends:
> 1. **ElChat achieves better chat and instruction-following abilities and source language performance than CV**; This generally holds across the board for these critical functionalities. Specifically, ElChat surpasses CV in every IFEval and GSM8K evaluation, regardless of language or model. Furthermore, ElChat consistently beats CV on MT-Bench for Qwen2.5 and remains competitive with Llama 3.1, experiencing only a 0.12 point drop on average.
> 2. **In target languages and safety tasks, ElChat is competitive and often more robust (i.e. consistently outperforming the source chat model) than CV.** We acknowledge that CV sometimes outperforms ElChat in some target language and safety tasks, as we explicitly reported in the second paragraph of Subsection 5.1 on page 6 for safety tasks.
>
> Crucially, the core motivation of ElChat is to overcome the reliance on access to a base model from the same family, a fundamental constraint of CV. ElChat provides a more flexible and broadly applicable solution, achieving strong and robust performance without this prerequisite.
>
> In the revised manuscript, we will expand our discussion on the comparison between ElChat and CV for target language tasks (e.g., GMMLU) in Subsection 5.2, providing granular insights into the performance of ElChat and CV across all evaluated tasks, clearly highlighting where each method excels or shows limitations.
>
> ---
>
> **On the rationale behind the primary use of SLERP**: As mentioned in footnote 5 on page 4, we also confirm that linear merging performs comparably. However, we primarily utilize SLERP because it offers more fine-grained control for merging, even if it is not always empirically superior across all tasks (Goddard et al., 2024). Our additional linear merging experiments indeed aim to confirm the overall robustness of our merging approach regardless of the specific interpolation method, which also helps reduce experimental complexity. We will elaborate on this rationale within Section 3 of the revised manuscript.
>
> Arcee’s MergeKit: A Toolkit for Merging Large Language Models (Goddard et al., EMNLP 2024)

---

### Author Response · Authors · 2025-07-07
**Revision**

Dear Reviewers nw3R, f5vZ, and BFxi,

We sincerely thank you for your constructive and valuable feedback. We have extensively addressed the points raised and incorporated significant revisions into the manuscript, highlighted in blue for easy identification. Below, we outline our changes:

* **On the distinction between a chat and base model** (Reviewer nw3R): We have clarified the distinction between chat and base models in both the Figure 1 caption and the second paragraph on page 1.

* **On the novelty of ElChat** (Reviewers nw3R, f5vZ, and BFxi): We have explicitly stated that ElChat's novelty stems from its strategic combination of existing techniques to address specific challenges, rather than its individual components (Section 3, second paragraph).

* **On the rationale behind the primary use of SLERP** (Reviewer BFxi): We have expanded the rationale behind the primary use of SLERP over linear merging in Footnote 5 on page 5.

* **On the methodological description** (Reviewers f5vZ and nw3R): Following suggestions from Reviewers f5vZ and nw3R, we have substantially expanded our methodology section to improve clarity. This includes a new conceptual figure (Figure 2) illustrating ElChat's processes and a step-by-step description of each involved process.

* **On the TruthfulQA, ToxiGen, and ImplicitHate evaluation** (Reviewer f5vZ): We have now explicitly mentioned that these evaluation uses target language translated data following Cahyawijaya et al. (2024) in the first paragraph on page 7.

* **On the insufficient & additional experimentation** (Reviewers f5vZ and BFxi): Per suggestions from Reviewers f5vZ and BFxi, we have conducted the following additional experiments and incorporated their results:

    * **Qwen3 14B** (Reviewer f5vZ): We have experimented with Qwen3 14B as additional analysis. Due to computational resource constraints, we examine Amharic, Bengali, and Telugu. The results and analysis are available in Appendix C.3. We generally observe a similar trend in Qwen3 compared to Qwen2.5 and Llama 3.1, confirming the efficacy of ElChat. Nonetheless, as Bengali and Telugu are now officially supported by Qwen3, the corresponding target language improvements are hardly seen (except for inference speedups).

    * **BBH** (Reviewer BFxi): We have added the results on BBH for all models and languages (Figure 4, Tables 11, 12, and 13). The overall trends are consistent with those in other English-centric benchmarks.

    * **AlpacaEval v2.0** (Reviewer BFxi): We have added the results on AlpacaEval for selected languages: Amharic, Bengali, and Telugu, due to computational constraints. The corresponding results are available in Table 10. We also have added a brief mention in the third paragraph of Chat and Instruction-following on page 9.

* **On the discussion comparing ElChat and Chat Vector** (Reviewer BFxi): Following our response to Reviewer BFx, we have made it clearer when CV outperforms ElChat and vice versa for target language tasks in the third paragraph of 5.2.

* **On the ethical consideration discussion** (Reviewer BFxi): Following the suggestions from Reviewer BFxi, we have expanded our discussion on the risks related to the deployment of adapted chat LMs in the Ethical Considerations section.

We hope this revision effectively addresses your insightful comments, resulting in a much improved and more robust version of our manuscript.

---

> ### Author Response · Authors · 2025-08-21
> **Final Recommendation for Paper4876**
>
> Dear Reviewers nw3R, f5vZ, and BFxi,
>
> Thank you again for your valuable feedback on our paper.
>
> As the author-reviewer discussion has now been open for nearly two months, we are keen to bring the process to a conclusion. To that end, we would be grateful if you could submit your final recommendation in the coming days.
>
> If you have any additional points you believe would strengthen the work for the final version, we would welcome those as well.
>
> We look forward to receiving your final assessment.
>
> Sincerely,
>
> The Authors of Paper4876

---

### Decision · Action_Editor_gBrF · 2025-09-14

**Recommendation:** Accept as is

**Additional Comments:**

In the final version, please explain more about the definition of the so call "chat model" in the paper, and explain more about the motivation about VE as most LLMs have a large vocabulary.

**Audience:**

Yes

**Audience Explanation:**

Yes. The problem of adapting LLMs and chat models to low-resource languages is timely, important, and highly relevant to the TMLR audience. The paper provides a practical method (ElChat) that demonstrates scalability, robustness, and measurable benefits for fairness in LLM applications. Researchers and practitioners working on multilingual LLMs, instruction-tuning, and efficient adaptation strategies will find these findings valuable.

**Claims And Evidence:**

Yes

**Claims Explanation:**

Yes. The authors have provided extensive empirical evidence across multiple benchmarks, including AlpacaEval v2.0 and Big-Bench Hard (BBH), and further strengthened the paper with experiments on the newer Qwen3 14B model. The results consistently corroborate the paper’s claims regarding improved instruction-following and reasoning abilities in low-resource language adaptation. The ablation analysis further supports the methodological contributions, showing the necessity of both model merging and weight copying. Overall, the claims are well supported by rigorous experiments and clear explanations.

---

> ### Author Response · Authors · 2025-09-29
> **Camera-ready submission**
>
> Dear Action Editor Ruoyu Sun,
>
> Thank you very much for accepting our paper for publication in TMLR. We are grateful for your time and insightful comments, which have helped us improve the manuscript.
>
> We are writing to confirm that we have now submitted the camera-ready version. In the final manuscript, we have addressed your final suggestions:
>
> * On the definition of the "chat model": The camera-ready version explains its definitions in Abstract (L3-L4), the second paragraph of the Introduction, and the caption of Figure 1.
>
> * On the motivation for VE against LLMs with large vocabularies: The camera-ready version explicitly mentions that (i) recent LLMs often have large vocabularies (e.g., 152K for Qwen2.5), but (ii) they still require substantially more inference steps for underrepresented languages like Amharic than their high-resource counterparts (e.g., English), necessitating the use of VE to mitigate this issue and achieve inference speedups.
>
> Thank you once again for your guidance throughout the review process. We are delighted to have our work published in TMLR.
>
> Best regards,
>
> Authors of Submission 4876